# T-bet+ B cells are activated by and control endogenous retroviruses through TLR-dependent mechanisms

Eileen Rauch[1,11], Timm Amendt[1,12], Aleksandra Lopez Krol[1], Fabian B. Lang[1], Vincent Linse[1], Michelle Hohmann[1,13], Ann-Christin Keim[1], Susanne Kreutzer[2], Kevin Kawengian[1], Malte Buchholz[3], Philipp Duschner[1], Saskia Grauer[1], Barbara Schnierle[4], Andreas Ruhl[1,14], Ingo Burtscher[5], Sonja Dehnert[1], Chege Kuria[6], Alexandra Kupke[7], Stephanie Paul[1], Thomas Liehr[8], Marcus Lechner[9], Markus Schnare[1], Andreas Kaufmann[1], Magdalena Huber[10], Thomas H. Winkler[6], Stefan Bauer[1] & Philipp Yu[1] ✉

Endogenous retroviruses (ERVs) are an integral part of the mammalian genome. The role of immune control of ERVs in general is poorly defined as is their function as anti-cancer immune targets or drivers of autoimmune disease. Here, we generate mouse-strains where Moloney-Murine Leukemia Virus tagged with GFP (ERV-GFP) infected the mouse germline. This enables us to analyze the role of genetic, epigenetic and cell intrinsic restriction factors in ERV activation and control. We identify an autoreactive B cell response against the neo-self/ERV antigen GFP as a key mechanism of ERV control. Hallmarks of this response are spontaneous ERV-GFP+ germinal center formation, elevated serum IFN-γ levels and a dependency on Age-associated B cells (ABCs) a subclass of T-bet+ memory B cells. Impairment of IgM B cell receptor-signal in nucleic-acid sensing TLR-deficient mice contributes to defective ERV control. Although ERVs are a part of the genome they break immune tolerance, induce immune surveillance against ERV-derived self-antigens and shape the host immune response.

Endogenous retroviruses (ERVs) represent mostly inactivated, proviral DNA derived from ancient retroviral infections of the germline. They account for a staggering 8–10% of the genome[1,2]. A fair amount of information has been obtained about the genetic and epigenetic control of ERVs during embryonic development[3]. However, how ERV reactivation and replication is regulated postnatally is less well understood. This is of growing importance since human ERV transcription is induced by epigenetic modifier anti-cancer drugs[4,5]. It is still unclear whether these HERV-transcripts are potential tumor antigens[6]. Importantly, it was recently demonstrated that antibodies

[1]Institute of Immunology, Philipps-Universität Marburg, 35043 Marburg, Germany. [2]Max-Planck-Institute for Heart and Lung Research, 61231 Bad Nauheim, Germany. [3]Department of Gastroenterology, Endocrinology and Metabolism, and Core Facility Small Animal Multispectral and Ultrasound Imaging, Philipps-Universität Marburg, 35043 Marburg, Germany. [4]Department of Virology, Paul-Ehrlich-Institut, 63225 Langen, Germany. [5]Institute of Diabetes and Regeneration Research, Helmholtz Zentrum München, 85764 Neuherberg, Germany. [6]Nikolaus-Fiebiger Center for Molecular Medicine, Friedrich-Alexander Universität Erlangen-Nürnberg, 91054 Erlangen, Germany. [7]Institute of Virology, Philipps-Universität Marburg, 35043 Marburg, Germany. [8]Jena University Hospital, Friedrich Schiller University, Institute of Human Genetics, 07747 Jena, Germany. [9]Center for Synthetic Microbiology, Philipps-Universität Marburg, 35043 Marburg, Germany. [10]Institute of Sytems Immunology, Center for Tumor and Immunobiology, Philipps-Universität Marburg, 35043 Marburg, Germany. [11]Present address: CSL Behring Innovation GmbH, Emil-von-Behring-Str. 76, 35041 Marburg, Germany. [12]Present address: The Francis Crick Institute, NW1 1AT London, UK. [13]Present address: Apollo Ventures Holding GmbH, 20457 Hamburg, Germany. [14]Present address: Department of Infection Biology, University Hospital Erlangen, 91054 Erlangen, Germany. ✉e-mail: Philipp.Yu@staff.uni-marburg.de

against ERVs promote lung cancer immunotherapy[7] in line with early concepts of ERV-immune surveillance mediated by neutralizing anti-ERV antibodies[8].

The importance of innate immune sensors was demonstrated by the finding that Toll-like receptor 7 (TLR7) controls exogenous murine leukemia virus (MuLV)-infections by induction of neutralizing antibodies[9]. Uncontrolled ERV-replication in B cell deficient mice confirmed the importance of functional B cell activation by ssRNA or retroviruses[10]. It is still a matter of debate whether increased exposure to ERV-activating bacterial components e.g. LPS, or the absence of neutralizing antibodies causes reactivation of ERVs in TLR- and B-cell-deficient mice[10–12]. Moreover, the fundamental question whether ERVs are self or non-self[13] could not be addressed because of the numerous immunologically closely related natural ERV sequences in the mouse genome[14].

In this work, we address this problem by extending the pioneering work of Rudolf Jänisch[15] and introduce a replication competent Moloney-MuLV (Mo-MuLV) into the mouse germline. GFP is inserted into the envelope (Env) gene, making Env-GFP an integral part of the membrane protein of the new ERV[16]. The provirus encoded GFP is a bona fide reporter to track ERV-GFP reactivation. This is instrumental to demonstrate that ERV-GFPs are expressed neonatally and thereby allow GFP to act as a neo-*auto*antigen for the immune system. In contrast to other GFP-reporter mice only EGT-315 (ERV-GFP-Tagged-315) mice produce anti-GFP IgG. The anti ERV-GFP immune response shares characteristics with B cell autoimmune responses e.g. spontaneous germinal centers with deposition of GFP on follicular dendritic cells (FDCs). Most surprisingly ERV-GFP is able to induce elevated systemic levels of IFN-γ and increased numbers of T-bet⁺ B-cells dependent on nucleic-acid-specific TLRs. T-bet⁺ ABCs (Age-associated B cells) are antigen-experienced memory B cells (MBC) found during chronic virus infections or autoimmunity[17] and are dependent on TLR7 and IFN-γ signals[18]. We propose that during evolution the immune system developed into a crucial component of ERV control. Chronic reactivation leads to nucleic-acid sensing by TLRs which induce ERV-specific ABCs overcoming B cell tolerance mechanisms similar to autoreactive B cells.

## Results

### ERV-GFP provirus is neonatally reactivated and expressed in a fraction of immune- and gut-epithelia cells

To generate germline ERV-GFP mice we infected embryonic stem (ES) cell line V6.5f1 (C57BL/6×129/Sv) with pMOV-GFP-derived virus (Fig. 1a). ES cell DNA was tested positive for integration of provirus (Fig. 1b). In line with epigenetic inactivation of proviral Mo-MuLV sequences in ES cells we could not detect reactivation of the provirus as judged by lack of GFP mRNA and protein expression (Fig. 1c–e). However, mouse embryonic fibroblasts (MEF) used as feeders for ES cells were positive for ERV-GFP virus expression (Fig. 1d, e). This confirms the previously described mechanisms of inhibition for ERV-GFP in ES cells[19]. By blastocyst-injection we obtained the germline transmitting founder EGT-315 which was backcrossed to C57BL/6 background (the suffix B6 is used to identify mice with wild type background).

Detection of the GFP signal by flow cytometry of peripheral blood cells indicates transcription and translation of ERV-GFP (Fig. 1f, upper panel). To test replication competence and infectivity we isolated plasma to infect the mouse B cell line WEHI-231. After 7 days GFP was detectable in 5 out of 6 cell cultures incubated with plasma from different ERV-GFP expressing mice (Fig. 1f, lower panel) confirming the presence of infectious virus in EGT-315 mice.

Birth terminates the protection of the embryo *in utero* from the massive exposure of potential ERV-activating exogenous stimuli, e.g. microbiota and dietary activators[20]. This coincides with a not yet fully developed innate and adaptive immune system of the neonate[21]. To avoid cross placenta transfer of maternal antibodies to ERV-GFP we analyzed offspring from female C57BL/6 mated to heterozygous EGT-315 B6 males. In neonatal EGT-315 mice analyzed 1 week after birth we found between 15–1200 ERV-GFP⁺ cells/100.000 splenocytes in both EGT-315 B6 and EGT-315 Tlr7⁻/⁻ mice (Fig. 1g). This suggests that when the adaptive immune system is not yet developed ERV-GFPs are already expressed.

In adult mice cells from organs such as liver, brain and kidney were negative for ERV-GFP expression (Fig. S1a), demonstrating that ERV-GFP is not generally reactivated in all tissues and cell types. In accordance with lymphocyte-specific tropism of exogenous infections by Mo-MuLV[22] mice backcrossed to C57BL/6 background expressed ERV-GFP in cell-type specific manner in spleen and bone marrow (Fig. 1h, S1b). Only a fraction of B- and T lymphocytes and dendritic cells were ERV-GFP positive on the C57BL/6 background (generations F1-F3, Fig. 1h). At the beginning of the project we analyzed EGT-315 Tlr7⁻/⁻ mice because of TLR7's well established key role as suppressor of ERVs[23]. However, as we previously showed only about 62.5% of Tlr7⁻/⁻ mice but 100% of Tlr3⁻/⁻Tlr7⁻/⁻Tlr9⁻/⁻ display ERV reactivation[11] we reasoned that comparison of EGT-315 Tlr3⁻/⁻Tlr7⁻/⁻Tlr9⁻/⁻ and EGT-315 B6 is important to better understand ERV-GFP control in vivo. As expected, EGT-315 Tlr3⁻/⁻Tlr7⁻/⁻Tlr9⁻/⁻ mice display an increased percentage of GFP positive cells in the same cell populations compared to EGT-315 B6 mice (Fig. 1h). Ly6G⁺ granulocytes displayed the highest percentage of GFP-positive cells with up to 34.6% in bone marrow and 54% in spleen of EGT-315 Tlr3⁻/⁻Tlr7⁻/⁻Tlr9⁻/⁻ mice. Interestingly, ERV-GFP expression leads to increased incidences of T cell acute lymphatic leukemia (T-ALL) with a shift from late (in Tlr3⁻/⁻Tlr7⁻/⁻Tlr9⁻/⁻ mice) to early onset in EGT-315 Tlr3⁻/⁻Tlr7⁻/⁻Tlr9⁻/⁻ mice (Fig. S1a, c).

Most strikingly, the whole gut (Fig. S1a) and, in particular, in epithelium of the small intestine ERV-GFP⁺ individual crypts were identified in both EGT-315 genotypes (Fig. 1i). Surprisingly, in close proximity to ERV-GFP⁺ crypts, completely negative crypts are located (Fig. 1i). Indeed, expression is absent in 68.3% of the crypts of EGT-315 B6 mice and 45.7% in EGT-315 Tlr3⁻/⁻Tlr7⁻/⁻Tlr9⁻/⁻ (in generations F1-F4; Fig. 1i). Further backcrossing increases the GFP-negative percentage of crypts to 97.8% in EGT-315 B6 and 62.8% EGT-315 Tlr3⁻/⁻Tlr7⁻/⁻Tlr9⁻/⁻ (F8-F10) (see below), respectively. Also, cells located in Peyer's patches of the gut are completely GFP-negative (Fig. S1d).

Our observations indicate that ERV-GFP expression is organ-specific and tropism in vivo is not restricted to lymphocytes but extends to bone marrow Ly6G⁺ granulocytes and epithelia cells of certain crypts of the mouse gut.

### Genetic-, and host restriction factors control ERV-GFP in vivo

Protection against ERVs comprises a complex network of genetic factors, host restriction factors and epigenetic gene regulation mechanisms e.g. DNA methylation[19].

To dissect genetic and positional influences of the ERV-GFP insertion we employed a linker-based PCR to identify the integration site in the genome. We used Fluorescence In Situ Hybridization (FISH) (Fig. S2a–c) and whole genome sequencing (WGS) (Fig. S2d) to verify this. We analysed spleen DNA of a EGT-315 B6 mouse and confirmed the germline ERV-GFP integration on Chr.19. However, WGS showed a single additional integration on Chr.13 which presumable occured in somatic cells, while the germline ERV-GFP was inherited according to Mendel's laws (Supplementary Tables 2, 3).

The virus inserted as a single copy on chromosome 19 subband 19B in the large intron of Pcx (Pyruvate carboxylase) at position 4,560,500-4,671,780 [GRCm39/mm39] without substantially interfering with Pcx or Unc93b1 expression and allowing backcrossing to homozygous EGT-315 state (Fig. S2e–i).

We examined genetic and epigenetic aspects of ERV control in adult mice. While natural MuLV invaded the germ line eons ago, we were able to follow the expression dynamics for the first generations of

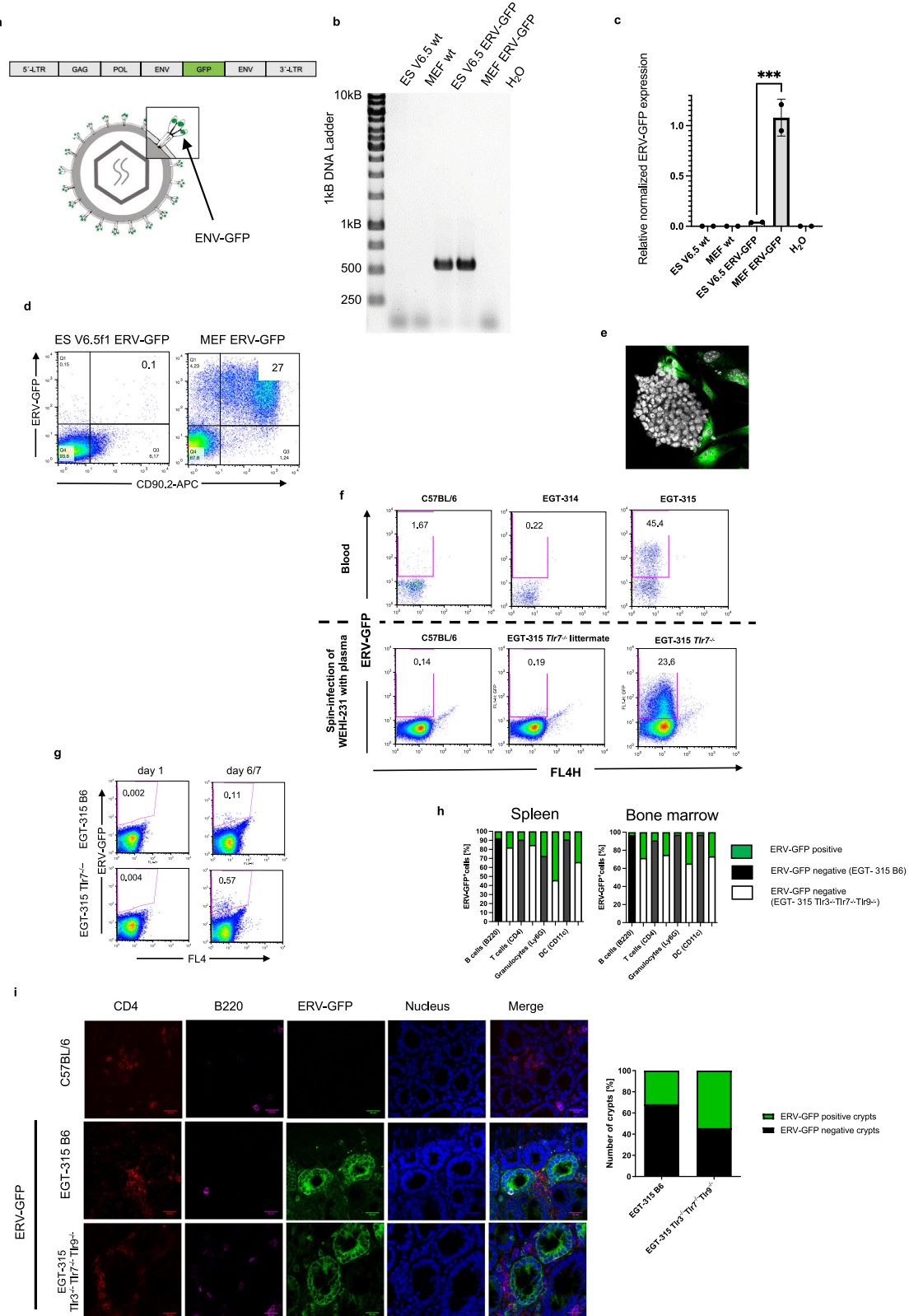

a new replication competent ERV. Backcrossing of EGT-315 mice to C57BL/6 (8 generations) and Tlr3$^{-/-}$Tlr7$^{-/-}$Tlr9$^{-/-}$ (11 generations) lead to a decline of ERV-GFP expression in peripheral blood (Fig. S2j). Notably, this is in accordance with the observation that complete control of natural ERVs is present in C57BL/6 mice[11]. In contrast, increased ERV-GFP expression is present in EGT-315 Tlr3$^{-/-}$Tlr7$^{-/-}$Tlr9$^{-/-}$ over at least 11 generations of backcrossing (Fig. S2j). Change of expression over the

relative short period is not due to loss of the integrated provirus (Fig. S2k).

We then addressed the question whether modifying genes from the 129/Sv background of the ES cell line V6.5f1 (C57BL/6 ×129/Sv) could explain reduced ERV-GFP expression in backcrosses to the C57BL/6 background. We mated EGT-315 B6 (F9) mice with low ERV-GFP expression with mice of the 129/Sv background creating progeny

**Fig. 1 | Generation and expression of ERV-GFP in the mouse strain EGT-315. a** Schematic genome and virus structure of ERV-GFP Mo-MuLV. Magnification: env-GFP fusion protein (green). **b** Detection of ERV-GFP by PCR in genomic DNA of embryonic stem (ES) cells and mouse embryonic fibroblasts (MEFs). **c** Expression of ERV-GFP RNA transcripts analyzed by Q-PCR in MEFs and ES cells. Single aliquots of uninfected (wt) and ERV-GFP infected ES cells and MEFs were tested. Duplicate measurements ES V6.5 ERV-GFP mean = 0.039 (technical replication), SD = 0,004 and MEF ERV-GFP, mean = 1.1, SD = 0.18 (technical replication). Statistics of technical replication with ordinary one-way ANOVA, ***$p$ = 0.0003. **d** ERV-GFP protein detection by flow cytometry. Anti-CD90.2 staining was used to differentiate ES cells from MEF feeder cells in co-culture. **e** Microphotograph of ERV-GFP infected ES-cell colony and MEFs. ERV-GFP (green; feeder cells) and nuclear counterstaining with DAPI (gray; all cells). **f** Flow cytometry for ERV-GFP expression in blood from 3-week old founder mouse EGT-315 backcrossed to C57BL/6 background and negative controls (C57BL/6 and EGT-314 non-transgenic founder, upper row). Example of in vitro infection assay of WEHI-231 cells by plasma of EGT-315 Tlr7$^{-/-}$ mice (genotype: Tlr3$^{+/-}$Tlr7$^{-/-}$Tlr9$^{+/-}$) (5 positive of $n$ = 6) and EGT-315 wild type Tlr7$^{-/-}$ littermates ($n$ = 3) and C57BL/6 ($n$ = 3) (lower row). **g** Representative flow cytometry of neonatal ERV-GFP activation in spleen, on day 1 and day 6 and 7 in EGT-315 C57BL/6 ($n$ = 0/2 on day 1 and 3/3 on day 6) and EGT-315-Tlr7$^{-/-}$ mice (genotype: Tlr3$^{+/-}$ Tlr7$^{-/-}$ Tlr9$^{+/-}$) ($n$ = 0/2 and 4/6 on day 7). **h** Flow cytometry of cell type-specific expression of ERV-GFP in cells of spleen and bone marrow of F1-4 EGT-315 B6 mice (black bars, $n$ = 5) as well as EGT-315-Tlr3$^{-/-}$Tlr7$^{-/-}$Tlr9$^{-/-}$ mice (white bars, $n$ = 5). Green indicates the percentage of ERV-GFP$^+$ cells relative to 100% of the cell type. EGT-315 B6 Spleen: B cells (mean = 7.6, SD = 4.3), T cells (mean = 9.2, SD = 7.5), Granulocytes (mean = 27, SD = 14.1), DCs (mean = 9, SD = 4.2). EGT-315 B6 bone marrow: B cells (mean = 3.2, SD = 2.4), T cells (mean = 8.9, SD = 9.3), Granulocytes (mean = 2.8, SD = 2.5), DCs (mean = 2.8, SD = 0.042). EGT-315 Tlr3$^{-/-}$Tlr7$^{-/-}$Tlr9$^{-/-}$ spleen: B cells (mean = 17.7, SD = 5.7), T cells (mean = 15, SD = 6.3), Granulocytes (mean = 54, SD = 21.2), DCs (mean = 34, SD = 16.8). EGT-315 Tlr3$^{-/-}$Tlr7$^{-/-}$Tlr9$^{-/-}$ bone marrow: B cells (mean = 28.5, SD = 13.6), T cells (mean = 25, SD = 0.31), Granulocytes (mean= 34.6, SD = 4.7), DCs (mean = 26.6, SD = 11.3). **i** Confocal microphotograph from small intestine of 2.5-month old mice F1-4; helper T cells (CD4), B cells (B220) and nuclei (DAPI). GFP signal indicates ERV-GFP expression. Intestines of 2.5-month old mice representative for C57BL/6 mice ($n$ = 4), EGT-315 B6 mice ($n$ = 4), and EGT-315 Tlr3$^{-/-}$Tlr7$^{-/-}$Tlr9$^{-/-}$ mice ($n$ = 6). Scale bar 20 µm. Right pannel, summary of ERV-GFP positive crypts (green) compared to negative crypts (black) from generations F1-4 EGT-315 B6 mice ($n$ = 4), and EGT-315 Tlr3$^{-/-}$Tlr7$^{-/-}$Tlr9$^{-/-}$ mice ($n$ = 5). EGT-315 B6 31,7% (83 GFP$^+$ crypts of 262) and EGT-315 Tlr3$^{-/-}$Tlr7$^{-/-}$Tlr9$^{-/-}$ mice 54,3% (139 GFP$^+$ crypts of 256). Source data are provided as a Source Data file.

with 50% C57BL/6 and 50% 129/Sv genetic background. This leads to a significant increase of ERV-GFP expression in blood cells (Fig. 2a) and suggests that loss of multiple permissive modifier genes from 129/Sv allels contribute to decreased ERV-GFP expression in backcrosses with C57BL/6.

To test whether epigenetic control of the ERV-GFP provirus is equally important in adult mice as in the embryonic stage we examined DNA-methylation status of the provirus. In ERV-GFP containing ES-DNA the GFP sequence of the proviral DNA is methylated while in ERV-GFP expressing B cell line WEHI-231 DNA-methylation is absent (Fig. 2b). To compare cells with and without virus expression we sorted GFP$^+$ versus GFP$^-$ immune cells from spleen and bone marrow of EGT-315 mice. However, we could not demonstrate a difference in methylation in the ERV-GFP region of the virus we analyzed (Fig. 2b). This suggests that epigenetic silencing of proviral ERV-GFPs in vivo may engage different mechanisms than DNA-methylation.

Seminal work of Stephen Goff's laboratory identified a molecular network of different host encoded restriction factors (HERFs) that block or prevent replication retroviruses[24]. Apolipoprotein B mRNA-editing catalytic polyprotein (Apobec) is one of the best understood factors which blocks retroviruses replication by mutating virus genomes through its cytidine deaminase activity[25]. Expression of ERV-GFP in EGT-315 on Tlr7$^{-/-}$ and Tlr3$^{-/-}$Tlr7$^{-/-}$Tlr9$^{-/-}$ background (Fig. 1h) indicates that mouse Apobec3G (and all other HERFs) are not able to completely prevent ERV-GFP replication in vivo.

However, hA3 (human Apobec3) added as a transgene (hA3Tg)[26] resulted in complete suppression of virus expression, judged by the lack of GFP signal (Fig. S3a). In line with hA3 function we could detect a reduction of ERV-GFP mRNA transcripts (Fig. 2c, left panel). More importantly, in EGT-315 Tlr3$^{-/-}$Tlr7$^{-/-}$Tlr9$^{-/-}$hA3 the transcripts displayed a high frequency of mutations in the GFP coding sequence (Fig. 2c, right panel). The main mechanism of ERV-GFP suppression might therefore involve introduction of incapacitating mutations which interrupt the replication cycle. This confirms recent data on human Apobec function[23].

Together, DNA methylation of the proviral ERV-GFP sequence might not be the dominant epigenetic mechanism that supresses ERV-reactivation in adult mice but rather unknown modifier allels absent in 129/Sv background. Also, the rare occurrence of EGT-315 B6 mice displaying high ERV-GFP positive cell counts (>10%) points to additional regulatory mechanisms for ERVs. Our model underscores the multilfaceted nature of retrovirus control mechanisms in vivo and suggests that they are not absolutely effective thereby allowing sufficient ERV expression for recognition by the innate immune system and B cells.

## Env-GFP membrane autoantigen must be part of the virus particle to induce Tlr7-dependent autoreactive anti-GFP IgG

Expression of IgG antibodies against self-antigens and in particular membrane autoantigens[27] is tightly regulated to avoid autoimmunity[28,29]. Notwithstanding, adult C57BL/6 mice harbor naturally occurring antibodies against ERVs[8] including their membrane envelope glycoprotein Env[11].

Here, GFP was inserted into the proline-rich region of the Env gene generating a fusion protein which is fluorescent while retaining infectivity of the virus (Fig. 1a, f). The activation of the provirus DNA of EGT-315 results in deposition of the Env-GFP fusion protein on the surface of cells (Figs. 1a, 3d, S3b). After budding ERV-GFP virus particles display Env-GFP on the the virus surface. This is probably the antigen that is recognized by antigen specific B cells and leads to production of GFP specific IgG (Fig. 3a) while all other GFP-expressing reporter mouse lines were negative for anti-GFP antibodies (Fig. 3a). Stratification of EGT-315 B6 mice into ERV-GFP high (>40%), low (2–20%) and negative (0–0.5%) expressors shows inverse correlation of GFP expression to the level of anti-GFP antibody titers (Fig. 3b). EGT-315 Tlr7$^{-/-}$ mice and consistently EGT-315 Tlr3$^{-/-}$Tlr7$^{-/-}$Tlr9$^{-/-}$ mice expressed ERV-GFP but no anti-GFP antibodies (Fig. 3b, c, f).

Importantly, we could not detect anti-GFP IgG antibodies in the mouse line designated EZGT where GFP was not expressed as an integral part of the virus particle but rather linked to the virus genome via an IRES element, ERV-IRES-GFP tagged Mo-MuLV[30] (Fig. 3e, f).

Therefore, only proteins expressed as integral part of the virus particle are capable of eliciting a specific IgG response, regardless of whether they are encoded as part of the mouse genome or represent a membrane protein.

## The autoreactive anti-ERV-GFP antibody response is characterized by Tlr-dependent spontaneous germinal center formation and its dependency on T-bet

Having established that infectious ERV-GFP and anti-GFP antibodies are expressed in vivo we characterized the ERV-GFP specific B cell response.

To this end we employed ELISpot assays (Fig. S3c) and generated hybridomas from EGT-315 B6 splenocytes to identify anti-GFP IgG producing B cells. Both experiments revealed the presence of anti-GFP IgG producing B cells in spleens of EGT-315 B6 mice with the capacity to block de novo infection in vitro (Fig. S3d, e).

To test whether in vivo activation of these B cells involves germinal center (GC)-reactions[31] we analyzed spleens of EGT-315 mice by immunohistochemistry. We could identify GCs in spleens of EGT-315 B6 mice which were strongly positive for ERV-GFP virus antigen. The

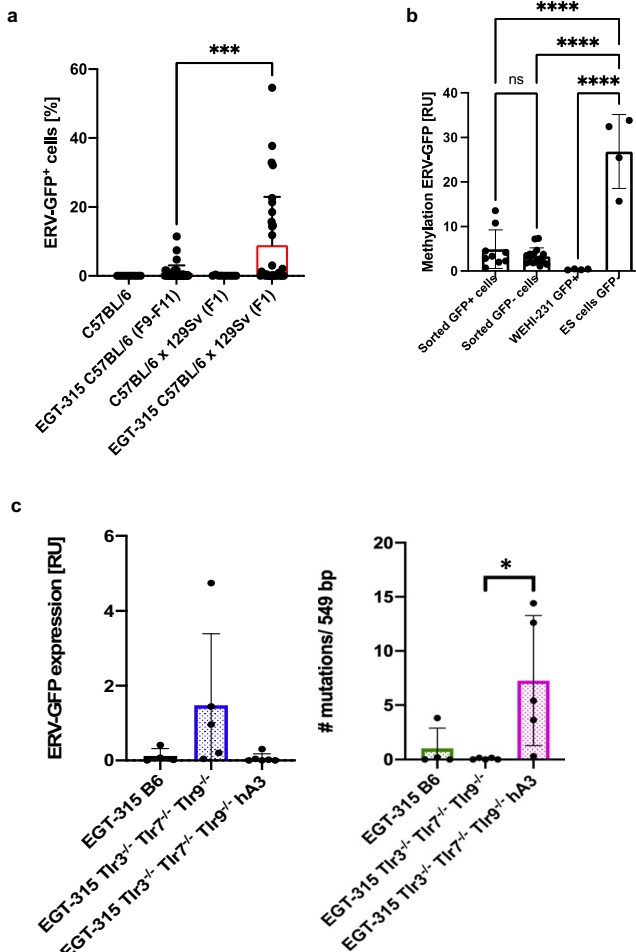

**Fig. 2 | ERV-GFP expression in vivo is influenced by modifier genes from the 129/Sv genetic background but not by epigenetic DNA methylation and suppressed by human cell intrinsic restriction factor Apobec3G (hA3). a** Reintroduction of 129/Sv genes by backcross of EGT-315 B6 (F9-F11) (mean = 0.79, SD = 2.23, *n* = 40) with 129/Sv increases ERV-GFP expression in EGT-315 B6 x 129Sv (F1) mice (mean = 8.91, SD = 14.0, *n* = 32). C57BL/6 (mean = 0, SD = 0, *n* = 9) and (129/Sv x C57BL/6) F1 wt littermates (mean = 0.063, SD = 0.122, *n* = 9). Red bars show mean. Flowcytometry of blood leukocytes at 3 weeks of age. ***p = 0.0008. Ordinary one-way ANOVA Tukey´s multiple comparsion test. **b** Epigenetic DNA methylation status of a 549 bp DNA fragment of ERV-GFP in GFP positive (+) vs GFP negative (-) cells isolated from spleen cells of EGT-315 mice. DNA was digested with methylation sensitive HpaII (recognition site CCGG) and normalized with Plcg2 specific PCR w/o HpaII recognition site. ES-cells are ERV-GFP containing but GFP-expression negative and ERV-GFP infected WEHI-231 cells strongly express GFP. From individual mice sorted GFP⁺ cells (mean = 4.91, SD = 4.32, *n* = 9), sorted GFP⁻ cells (mean = 3.3, SD = 1.9, *n* = 15), WEHI-231 GFP⁺ (mean = 0.37, SD = 0.12, *n* = 4), ES cells GFP⁻ (mean = 26.9, SD = 8.3, *n* = 4). Sorted GFP⁺ cells vs Sorted GFP⁻ cells ⁿˢP = 0.752. Sorted GFP⁺ cells vs ES cells GFP⁻ ****P < 0.0001. Sorted GFP⁻ cells vs ES cells GFP⁻ ****P < 0.0001. WEHI-231 GFP⁺ vs ES cells GFP⁻****P < 0.0001. Ordinary one-way ANOVA Tukey´s multiple comparsion test. Summary of 3 experiments. **c** Q-PCR of RNA expression of the GFP and Env region (549 bp) of ERV-GFP transcribed in spleen of EGT-315 mice (left panel). Sequence analysis of RNA expression analysis by RT-PCR-transcripts of EGT-315 mice with and without human Apobec3 (hA3Tg). Each dot depicts the mean number of mutations of 4–8 clones sequenced per individual mouse of the different genotypes (right panel). EGT-315 B6 (green bar, mean = 1, SD = 1.9, *n* = 4), EGT-315 Tlr3⁻/⁻ Tlr7⁻/⁻ Tlr9⁻/⁻ (blue bar, mean = 0.06, SD = 0.09, *n* = 5), EGT-315 Tlr3⁻/⁻ Tlr7⁻/⁻ Tlr9⁻/⁻ hA3 (purple bar, mean = 7.3, SD = 6.0, *n* = 5). Statistics of mutations EGT-315 Tlr3⁻/⁻ Tlr7⁻/⁻ Tlr9⁻/⁻ versus EGT-315 Tlr3⁻/⁻ Tlr7⁻/⁻ Tlr9⁻/⁻ hA3 *p = 0.028 by Ordinary one-way ANOVA Tukey´s multiple comparsion test. Source data are provided as a Source Data file.

distribution of the ERV-GFP signal was either a scattered network-like pattern in the center of the B cell follicle (Fig. 4a, left panels) or located as iccosomes in the FDC network of the GC light zone (Fig. 4a, right panels). Trapping of ERV-GFP virus antigen with anti-GFP IgG in immune complexes on the surface of FDC networks of GCs was corroborated by positive staining with FDC marker (CD21/35), anti-mouse IgG and peanut-agglutinin (PNA) (Fig. S3f). EGT-315 Tlr3⁻/⁻Tlr7⁻/⁻Tlr9⁻/⁻ mice display dispersed ERV-GFP⁺ cells and B cell follicles which were not organized in GCs (Fig. 4a). Interestingly, young EGT-315 B6 mice (1–2 month) showed a higher percentage of GFP⁺ B cell follicles (7.5%) which suggests that at this stage the initial immune response against ERV-GFP ensues. However, 2% of GFP⁺ B cell follicles in mice at 5 month of age suggests that constant re-emerging of ERV-GFP is sensed by the immune system (Fig. 4b).The GC B cell marker GL7 was most strongly expressed in EGT-315 B6 mice (Fig. 4c).

This suggests that nucleic acid-sensing (NAS)-TLRs recognize the virus and drive ERV-GFP-specific B cells in a bona fide GC reaction. This corresponds to spontaneous GC activation described previously[32] but in our model the spontaneous autoreactive GC B cell response can be traced back to ERV-GFP reactivation.

These results prompted us to look for differences of specific mRNA gene expression in splenic B cells. In EGT-315 B6 we were not able to detect the expression of both ERV-GFP and natural ERVs (MuLV) by RT-QPCR whereas both types of ERVs where strongly expressed in EGT-315 Tlr3⁻/⁻Tlr7⁻/⁻Tlr9⁻/⁻ (Fig. 4d, and see below). An explanation is that during B cell purification the ERV-GFP-IgG immune complex trapped on FDCs is lost while the higher percentage of ERV-GFP expressing B cells (Fig. 1h) of EGT-315 Tlr3⁻/⁻Tlr7⁻/⁻Tlr9⁻/⁻ mice leads to mRNA detection of the virus.

The heightened GC and anti-ERV-GFP antibody response in EGT-315 B6 is reflected by an increase in activation-induced cytidine deaminase (AID) and T-bet mRNA expression (Fig. 4d). The latter is probably caused by the increase in T-bet⁺ B cell frequency not present in other genotypes (Fig. 5a). Based on flow cytometry panels established by different laboratories[33,34] we used CD19⁺,CD21/35⁻, CD11b⁺/⁻, CD23⁻, CD11c⁺ with strong T-bet expression as markers to identify bona fide ABC cells (Fig. 5b). Analysis of EGT-315 B6 splenocytes suggests that the ABC population is completely devoid of GL-7⁺ GC cells (Fig. 5c). This excludes the possibility of germinal center B cell (GCB) involvement within the T-bet⁺ B cell compartment. Thus, mere insertion of a single copy of ERV-GFP into the genome of the non-autoimmune background C57BL/6 results in upregulation of ABCs.

This prompted us to investigate prerequisites and effects of elevated T-bet⁺ ABC numbers[17]. IFN-γ and signals via TLR7 and TLR9 together with chromatin/apoptotic debris or BCR engagement by microbial products have been identified as key inducers of T-bet⁺ B cells[17]. Here, we found that the insertion of ERV-GFP into the mouse genome leads to increased IFN-γ levels in EGT-315 mice (Fig. 5d). While C57BL/6 and Tlr3⁻/⁻Tlr7⁻/⁻Tlr9⁻/⁻ mice show intermediate levels, EGT-315 Tlr3⁻/⁻Tlr7⁻/⁻Tlr9⁻/⁻ mice display dramatically reduced systemic IFN-γ levels (Fig. 5d). At present the source of IFN-γ has not been identified. Analysis of mitogen stimulated CD4⁺, CD8⁺ T cells or NK cells derived from EGT-315 mice revealed no evidence that these cell types are the source for the hightened IFN-γ response (Fig. S4b). Also, splenic B cells did not show increased autocrine IFN-γ or STAT-1 mRNA upregulation (Fig. S4a).

The antibody response of T-bet⁺ B cells is dominated by the IgG2ₐ/c isotype[17]. Consistent with this and the elevated IFN-γ levels, we detected increased total and ERV-GFP-specific IgG2c levels only in EGT-315 B6 mice (Figs. 5e, S4c). To verify the role of T-bet in B cell mediated immunsurveillance of ERVs we generated EGT-315 B6 T-bet⁻/⁻ mice and compared the expression of ERV-GFP in blood and gut crypts in these mice to EGT-315 mice on B6 and Tlr3⁻/⁻Tlr7⁻/⁻Tlr9⁻/⁻ background. EGT-315 B6 mice showed no ERV-GFP⁺ cells in blood and the lowest number of GFP⁺ crypts (2.2%) (Fig. 5f, left panel, 5g). EGT-315 Tlr3⁻/⁻Tlr7⁻/⁻Tlr9⁻/⁻

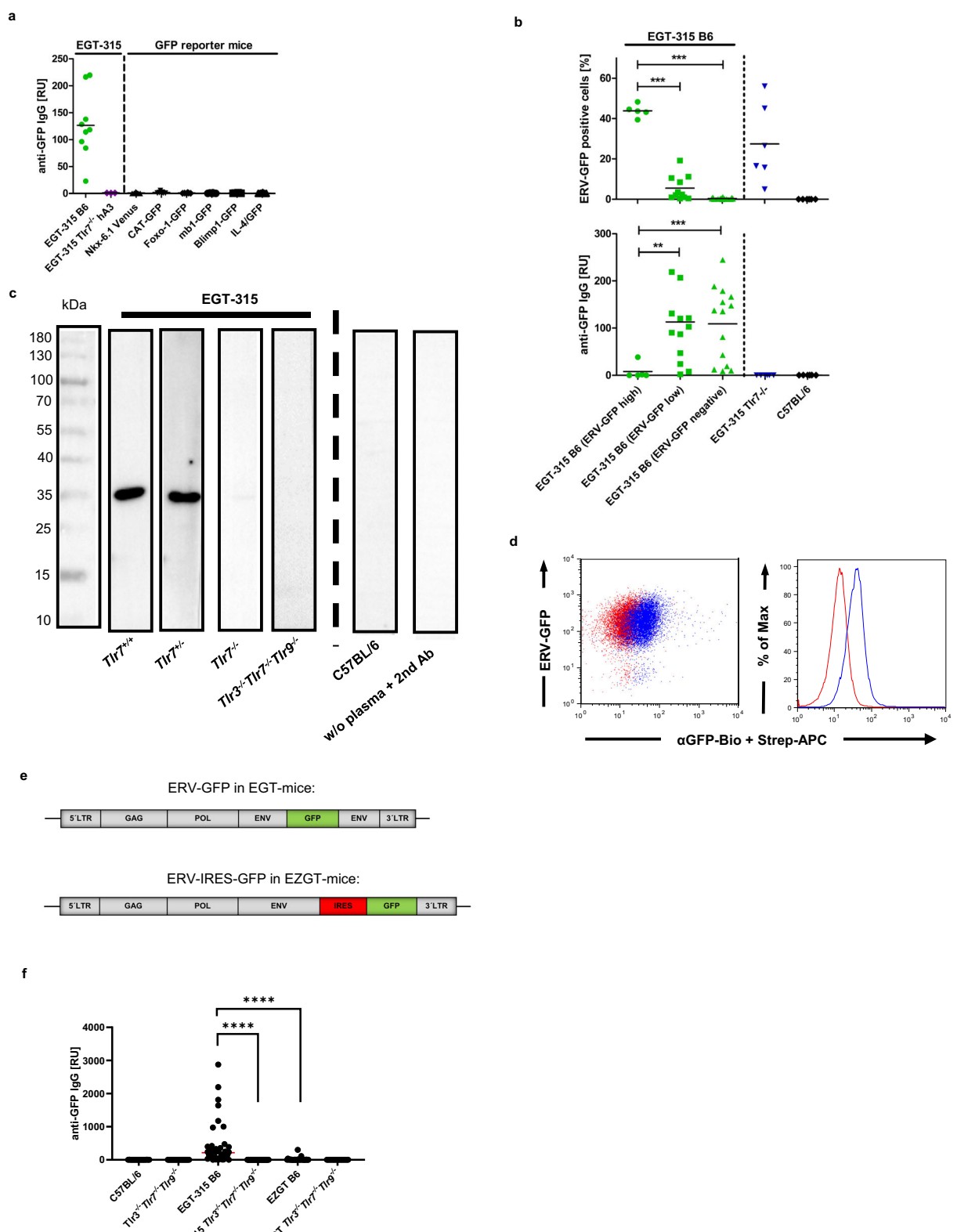

displayed ERV-GFP expression in 2.9% of PBMCs in 14/17 mice and crypts (37.2%). In 10 out of 26 EGT-315 B6 T-bet$^{-/-}$ mice (38.5%) we detected a high percentage of ERV-GFP$^+$ cells (>10%) of the PBMCs (Fig. 5f, left panel). Furthermore, in adult EGT-315 B6 T-bet$^{-/-}$ mice the number of ERV-GFP$^+$ crypts (12.8% GFP$^+$) is higher than in EGT-315 B6 but lower than in EGT-315 Tlr3$^{-/-}$Tlr7$^{-/-}$Tlr9$^{-/-}$ (Fig. 5g). The anti-GFP IgG and IgG2c titers of EGT-315 B6 T-bet$^{-/-}$ are clearly reduced in

comparison to EGT-315 B6 mice but not completely absent like in EGT-315 Tlr3$^{-/-}$Tlr7$^{-/-}$Tlr9$^{-/-}$ mice (Fig. 5f, middle and right panel). Nevertheless and surprsingly, EGT-315 T-bet$^{-/-}$ mice suffer from the highest T-ALL frequency probably due to ERV(-GFP)-viremia driven insertional mutagenesis[11] (Figs. 5h, S4d).

Therefore, in EGT-315 B6 mice low expression of ERV-GFP virus together with increase in ERV-GFP specific IgG2c, IFN-γ levels and

**Fig. 3 | Properties of the env-GFP membrane autoantigen that lead to TLR-dependent anti-GFP IgG response. a** Comparison of anti-GFP IgG antibody response measured by GFP-specific ELISA for EGT-315 B6 (green dots, $n = 9$) vs different GFP reporter mice, Nkx6.1 Venus ($n = 9$), CAT-GFP ($n = 7$), Foxo-1-GFP ($n = 6$), mb1-GFP ($n = 6$), Blimp1-GFP ($n = 6$), IL4/GFP ($n = 8$) and EGT-315 $Tlr3^{-/-}Tlr7^{-/-}Tlr9^{-/-}$-hA3 transgenic mice ($n = 3$). **b** Stratification of ERV-GFP expression and anti-GFP IgG production. Upper panel: flow cytometry of ERV-GFP expression as % positive peripheral blood cells per mouse. EGT-315 B6 mice (green, total $n = 30$) were subdivided into three groups: high percentage of ERV-GFP positive cells (green dots, $n = 5$), low (green squares, $n = 11$) and negative (green triangles, $n = 14$). EGT-315 $Tlr7^{-/-}$ (genotype: Tlr3$^{+/-}$Tlr7$^{-/-}$Tlr9$^{+/-}$) (blue triangles, $n = 6$) and C57BL/6 (black rhombuses, $n = 6$). Lower panel: corresponding anti-GFP IgG titers measured by GFP-specific ELISA. Statistics by two-tailed unpaired $t$-test with Welch´s correction, upper pannel GFP: EGT-315 B6 ERV-GFP$^{high}$ vs ERV-GFP$^{low}$ ***$P < 0,0001$; EGT-315 B6 ERV-GFP$^{high}$ vs. ERV-GFP$^{negative}$ ****$P < 0,0001$. lower pannel anti-GFP IgG: EGT-315 B6 ERV-GFP$^{high}$ vs ERV-GFP$^{low}$ **$P = 0,0013$; EGT-315 B6 ERV-GFP$^{high}$ vs ERV-GFP$^{negative}$ ***$P = 0,0004$. **c** Western blot of antibody production of EGT-315 mice against GFP protein blotted on stripes. Stripes were incubated with plasma from 2 to 4-month old mice. Visualization was done by anti-IgG-HRP detection. Representative for EGT-315 Tlr7$^{-/-}$ mice (genotype: Tlr3$^{+/-}$Tlr7$^{-/-}$Tlr9$^{+/-}$) ($n = 4$) EGT-315 Tlr7$^{+/-}$ mouse ($n = 1$), EGT-315 C57BL/6 mice ($n = 10$ positive from 11), EGT-315 Tlr3$^{-/-}$Tlr7$^{-/-}$Tlr9$^{-/-}$ ($n = 4$) and C57BL/6 mice ($n = 5$). **d** Env-GFP is expressed on the surface of cells. Flow cytometry of inhibitory anti-GFP mAb binding to ERV-GFP expressing cells. Biotinylated anti-GFP monoclonal Ab clone 174 was used to stain NIH-3T3 ERV-GFP. Anti-GFP mAb174 and Streptavidin-Alexa Fluor 633 (blue), only Streptavidin Alexa Fluor 633 (red) ($n = 3$). **e** Schematic genome structure of the recombinant Mo-MuLV used to generate EGT-315 (pMOV-GFP) and the EZGT-332/3 (pZAPM-GFP) mouse strains. **f** Comparison of anti-GFP IgG response of individual mice. Genotype: C57BL/6 (mean = 1.2, SD = 1.3, $n = 18$), Tlr3$^{-/-}$Tlr7$^{-/-}$Tlr9$^{-/-}$ (mean = 0.5, SD = 0.5, $n = 19$), EGT-315 B6 (mean = 452.4, SD = 672.9, $n = 37$), EGT-315 Tlr3$^{-/-}$Tlr7$^{-/-}$Tlr9$^{-/-}$ (mean = 0.48, SD = 0.48, $n = 27$), EZGT B6 (mean = 19, SD = 58.6, $n = 29$), EZGT Tlr3$^{-/-}$Tlr7$^{-/-}$Tlr9$^{-/-}$ (mean = 0.68, SD = 0.58, $n = 16$); EGT-315 B6 vs EGT-315 Tlr3$^{-/-}$Tlr7$^{-/-}$Tlr9$^{-/-}$ ****$P < 0.0001$; EGT-315 B6 vs EZGT B6 ****$P < 0.0001$. Ordinary one-way ANOVA Tukey´s multiple comparsion test. Summary of 3 experiments. Source data are provided as a Source Data file.

T-bet$^+$ B cells indicate that ERVs are constantly monitored and ultimately controlled by this B cell response. The effect of complete genetic ablation of T-bet supports the crucial role of T-bet in ERV control.

Of note, mice with Ighm$^{-/-}$ (B cell deficient) and Rag1$^{-/-}$ (T and B cell deficient) are unable to control ERVs while Tcra$^{-/-}$ (conventional T cell deficient), Tcrd$^{-/-}$ (γ/δ T cell deficient), and H2-A,E$^{-/-}$ (CD4$^+$ T cell deficient) are able to suppress their genome encoded ERVs[10]. These findings highlight the critical dependence of ERV control on B cells and B cell responses.

## Expression of ERV-GFP increases evolutionary-inherited ERV (MuLV) reactivation resulting in enhanced anti-MuLV antibody titers and autoimmune phenomena

We were interested in the influence of ERV-GFP expression on the reactivation of the genome encoded ERV in immune competent C57BL/6 and the B cell response against these inherited viruses. Furthermore, we set out to interrogate the capacity of ERV-GFPs to influence autoantibody formation.

In accordance with our observation in C57BL/6 mice[11] we could not detect MuLV-ERV mRNA transcripts in EGT-315 B6 mice by RT-QPCR (Fig. 6a). Surprisingly, in EGT-315 B6 mice which carry ERV-GFP in their genome the anti-MuLV (C57BL/6 ERV) IgG response was dramatically increased (10-fold) compared with C57BL/6 (Fig. 6b). This may be caused by induction of C57BL/6-genome encoded MuLVs/ERVs by ERV-GFP or reflect a crossreactvity of the immune response against common MuLV antigens shared between ERV-GFP (Mo-MuLV based) and the C57BL/6 ERVs.

Moreover, the anti-MuLV antibody response was completely absent in EGT-315 Tlr3$^{-/-}$Tlr7$^{-/-}$Tlr9$^{-/-}$, allowing significantly increased ERV-MuLV expression (Fig. 6a, b).

To examine whether ERV-GFP expression leads to autoantibody production we analyzed anti-DNA autoantibody formation. We detected increased titers of anti-DNA IgG in EGT-315 B6 mice (Fig. 6c). However, this was not reflected in increase of anti-nuclear antibodies in a HEp-2 indirect immunofluorescence assay (Fig. S4e, f). In contrast, reactivity against DNA and anti-nuclear antigens is completely absent in mice triple-deficient for Tlr3, Tlr7 and Tlr9 with and without ERV-GFP in both assays (Figs. 6c, S4e).

In summary, EGT-315 B6 mice show an increase in anti-ERV antibodies (MuLV and ERV-GFP), anti-DNA autoantibodies and expressed infectious ERV-GFP virus particles. This results in immune complex (IC) depositions containing ERV-GFP in the kidney glomeruli of EGT-315 B6 mice (Fig. 6d), but does not progress to kidney pathology e.g. glomerulonephritis (Fig. S4g). At the same time IC formation might be a way to sequester ERV-GFP in vivo and suppress its infectivity.

Interestingly, in TLR-deficient mice the absence of anti-ERV antibodies coincides with the lack of anti-nuclear and anti-DNA autoantibodies, suggesting a common pathway of B cell activation specific for nucleic acid driven autoreactivity and ERV control.

## Innate sensing of ERV-GFP has a systemic influence on the immune system

The idea that endogenous retroviruses shape the immune response has been discussed for some time[35].

Indeed we found that in comparison to C57BL/6 mice ERV-GFP containing EGT-315 B6 mice spontaneously produced more serum IFN-γ (Fig. 5b). This results in increased Th-1/IFN-γ−dependent IgG2c isotype in the serum of these mice (Fig. 5c). Consequently, genetic ablation of innate nucleic acid sensors in EGT-315 Tlr3$^{-/-}$Tlr7$^{-/-}$Tlr9$^{-/-}$ mice reduced systemic IFN-γ as well as IgG2c levels. A broader systemic immune imbalance caused by ERV expression in conjunction with TLR-deficiency results in elevated total IgE serum levels (Fig. 7a). As expected, high levels of IgE in EGT-315 Tlr3$^{-/-}$Tlr7$^{-/-}$Tlr9$^{-/-}$ mice leads to binding of IgE to FcεRI on basophils in vivo (Fig. 7b, S5a). C23-bound IgE inhibits the proteolytic cleaveage of CD23 by ADAM10[36] thereby stabilizing the low affinity receptor for IgE (CD23) on B cells. In line with elevated serum IgE levels in EGT-315 Tlr3$^{-/-}$Tlr7$^{-/-}$Tlr9$^{-/-}$ mice, we demonstrated increased CD23 expression on splenic B cells (Fig. S5b).

Since IgE expression is very tightly regulated we first tested if ERV-GFP virus is inducing antigen-specific IgE. However, GFP-reactivity of elevated IgE could not be verified (Fig. S4c). This is in accordance with a lack of ERV-GFP containing GCs in EGT-315 Tlr3$^{-/-}$Tlr7$^{-/-}$Tlr9$^{-/-}$ mice (Fig. 4a−c). It suggests either an extrafollicular IgE B cell response or the possibility that direct exposure or infection by ERV-GFP could activate B cells in an antigen-independent manner promoting class switching to IgE. We tested this in vitro by culturing B cells with purified ERV-GFP virus or 40LB[37] cells expressing ERV-GFP. No IgE induction could be observed while stimulation with IL-4 induced switching to IgE (Fig. S5c, d). Therefore, as a direct effect of the virus on B cells is unlikely we examined if a reduction of IFN-γ in EGT-315 Tlr3$^{-/-}$Tlr7$^{-/-}$Tlr9$^{-/-}$ mice is reflected in increased IL-4 levels. However, neither serum IL-4 nor frequency of IL-4$^+$CD4$^+$ T cells in vitro was elevated (Fig. S5e, f).

Overall, ERV-GFP expression, when properly sensed, leads to a global increase of serum IFN-γ, a cytokine that is capable of instructing anti-viral immune responses[38] and enabling the establishment of a Th-1 associated ERV-antigen specific IgG2c response. Impairment of ERV recognition by deletion of nucleic acid-specific TLRs reduces expression of IFN-γ, thereby allowing an increase in IgE due to an un-opposed but basically unchanged IL-4 level in vivo. Therefore, ERVs are able to globally influence the quality of the immune response depending on the genetic makeup of the host.

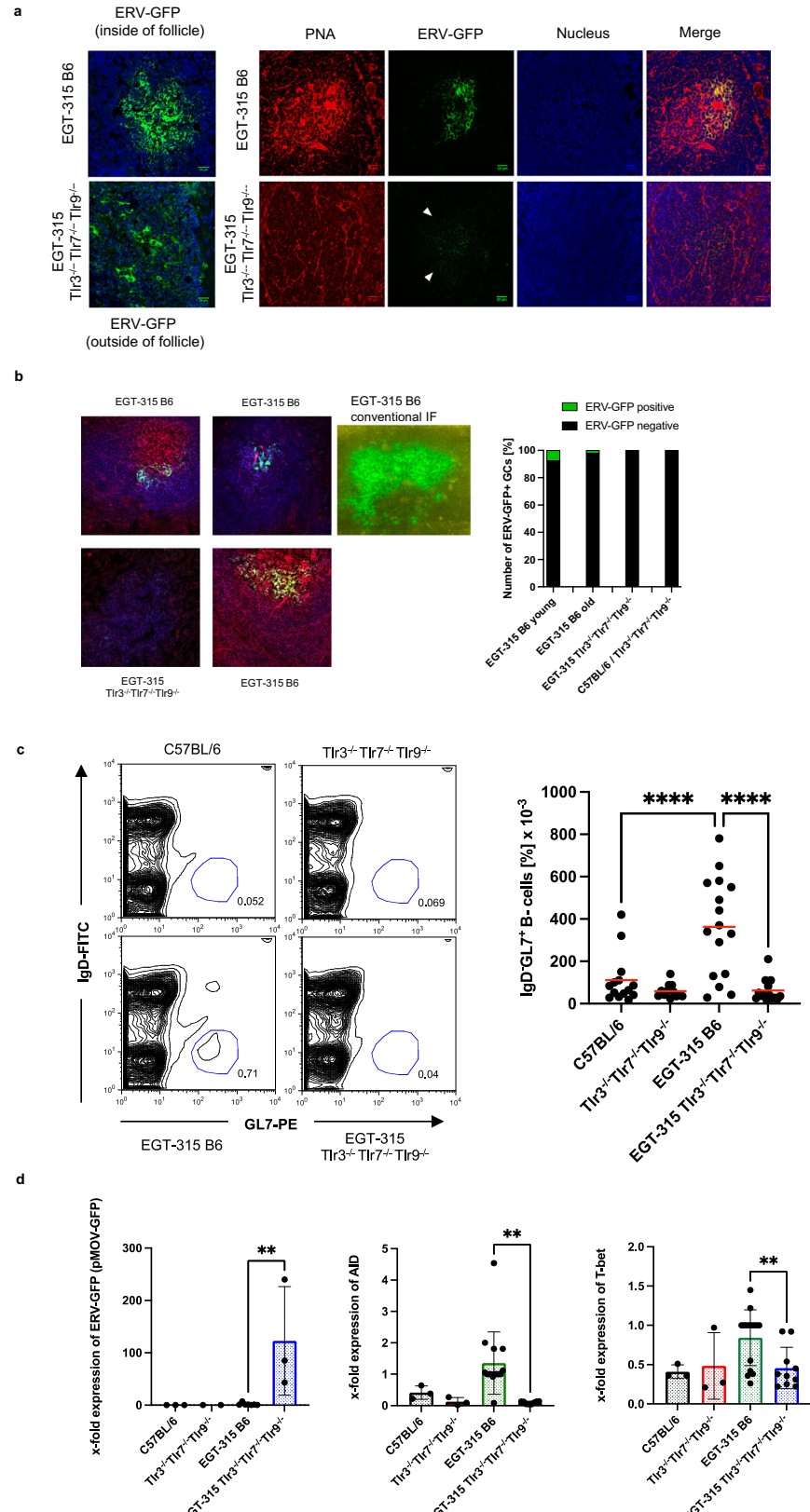

## Deficiency of nucleic acid-sensing TLRs impairs IgM-induced B cell survival/proliferation but induces abberant CD23 expression in vitro

Finally, we aimed to investigate the mechanism behind the complete lack of anti-ERV IgG and GC formation in nucleic acid-sensing (NAS) Tlr-deficient mice.

Tlr3[−/−]Tlr7[−/−]Tlr9[−/−]- and EGT-315 mice with and without NAS-Tlrs show normal B cell development including IgM and IgD expression-patterns corresponding to the stages of immature, transitional and mature splenic B cells (Fig. 8a, S5g). Additionally, similar IgM[+] memory B cell populations, using CD80 and CD73 markers, were found (Fig. S5h)[39]. The distribution of follicular vs. marginal zone B cells is

**Fig. 4 | The GFP-specific IgG response engages spontaneous germinal centers containing ERV-GFP immune complexes. a** GFP virus containing immune complexes on FDC network in B cell follicles and germinal centers of EGT-315 mice examined by confocal microphotography. Representative of spleens from EGT-315 B6 (n = 6) and EGT-315 Tlr3⁻/⁻Tlr7⁻/⁻Tlr9⁻/⁻ (n = 4) mice (scale bar 20 μm). Two left panels, ERV-GFP (green) and DAPI (blue). Right panels, spontaneous GCs stained with PNA-Alexa Fluor 647 (Peanut Agglutinin, red). Nuclei were stained with DAPI (blue). ERV-GFP (green). White triangles point to weak ERV-GFP signal in follicles. **b** ERV-GFP deposition in B cell follicles of EGT-315 B6 mice. Left, confocal microphotography representative of spleens stained with PNA-Alexa Fluor 647 (Peanut Agglutinin, red) and CD19-PE (blue). ERV-GFP is shown in green. Middle, microphotography of ERV-GFP deposition in standard immunofluorescence (IF) in green. Right, quantification of ERV-GFP (green) vs ERV-GFP negative (black) B cell follicles of individual mice: EGT-315 B6 young (positive 7.5%; 15 + /186-; n = 5; mean age = 1 m19d), EGT-315 B6 old (2%; 6 + /297-; n = 12; mean age = 5 m 24d), EGT-315 Tlr3⁻/⁻Tlr7⁻/⁻Tlr9⁻/⁻ (positive 0%; 0 + /348-; n = 10; mean age = 4m5d) and C57BL/6 Tlr3⁻/⁻Tlr7⁻/⁻Tlr9⁻/⁻ (positive 0%; 0 + /151-; n = 4; mean age = 8m14d). **c** Identification of IgD⁻GL7⁺ GC B cells in spleen. Left panel, exemplary flow cytometry of anti-IgD vs GL7 staining (Blue gate, IgD⁻GL7⁺). Right panel, statistical analysis of GL7⁺ B cells of C57BL/6 (mean = 111, SD = 117, n = 14) Tlr3⁻/⁻Tlr7⁻/⁻Tlr9⁻/⁻ (mean = 59, SD = 33, n = 12), EGT-315 B6 (mean = 363, SD = 232, n = 16), EGT-315 Tlr3⁻/⁻Tlr7⁻/⁻Tlr9⁻/⁻ (mean = 62, SD = 52, n = 15). C57BL/6 vs. EGT-315 B6 ****P < 0.0001; EGT-315 B6 vs. EGT-315 Tlr3⁻/⁻Tlr7⁻/⁻Tlr9⁻/⁻ ****P < 0.0001. Ordinary one-way ANOVA Tukey's multiple comparisons test. Summary of 8 experiments. **d** Q-PCR mRNA expression analysis of purified splenic B cells from individual mice. Values are x-fold expression relative to housekeeping gene actin. ERV-GFP (pMOV-GFP, left) C57BL/6 (black bar, mean = 0, SD = 0, n = 3), Tlr3⁻/⁻Tlr7⁻/⁻Tlr9⁻/⁻ (red bar, mean = 0, SD = 0, n = 2), EGT-315 B6 (green bar, mean = 1.7, SD = 2.3, n = 7), EGT-315 Tlr3⁻/⁻Tlr7⁻/⁻Tlr9⁻/⁻ (blue bar, mean = 122.7, SD = 84.6, n = 3), Summary of 7 experiments; **P = 0.0096; *AID* (middle) C57BL/6 (black bar, mean = 0.41, SD = 0.17, n = 3), Tlr3⁻/⁻Tlr7⁻/⁻Tlr9⁻/⁻ (red bar, mean = 0.13, SD = 0.1, n = 3), EGT-315 B6 (green bar, mean= 1.36, SD = 0.96, n = 15), EGT-315 Tlr3⁻/⁻Tlr7⁻/⁻Tlr9⁻/⁻ (blue bar, mean = 0.09, SD = 0.04, n = 8), Summary of 8 experiments; **P = 0.0019. *T-bet* (right) C57BL/6 (black bar, mean = 0.41, SD = 0.06, n = 3), Tlr3⁻/⁻Tlr7⁻/⁻Tlr9⁻/⁻ (red bar, mean = 0.48, SD = 0.34, n = 3), EGT-315 B6 (green bar, mean = 0.84, SD = 0.34, n = 15), EGT-315 Tlr3⁻/⁻Tlr7⁻/⁻Tlr9⁻/⁻ (blue bar, mean = 0.46, SD = 0.25, n = 10), Summary of 8 experiments, **P = 0.0074; For all Q-PCR statistics we used two-tailed unpaired t-tests. Source data are provided as a Source Data file.

identical (Fig. 8b), as is Ca²⁺-mobilization upon anti-IgM stimulation (Fig. S5i).

However, not only the anti-GFP IgG response is completely absent in EGT-315 Tlr3⁻/⁻Tlr7⁻/⁻Tlr9⁻/⁻ mice (Fig. 3a, c) but also the anti-GFP IgM response (Fig. 8c). This indicates that the molecular or cellular defect that leads to abolished immune control in vivo is already present at the IgM⁺ B cell stage. To test this we purified splenic B cells and stimulated them with anti-IgM polyclonal antibodies for 3 days to test for proliferation and viability using propidium iodide staining. The results show that Tlr3⁻/⁻Tlr7⁻/⁻Tlr9⁻/⁻ B cells do not react to anti-IgM B cell receptor crosslinking with survival/proliferation in vitro (Fig. 8d). Presence of ERV-GFP did not influence this phenotype (Fig. 8d). Proliferation to the TLR4 ligand LPS was comparable between all genotypes. As expected TLR7 and TLR9 ligands did not stimulate B cells from Tlr3⁻/⁻Tlr7⁻/⁻Tlr9⁻/⁻ mice. The IgM-BCR induced proliferation/survival signal in B cells from EGT-315 B6 and C57BL/6 mice is completely dependent on Bruton's tyrosine kinase (Btk). Interestingly, the residual survival signal in Tlr3⁻/⁻Tlr7⁻/⁻Tlr9⁻/⁻ B cells is dependent on Btk-signaling (Fig. 8e). The hypothesis that RNA or DNA released by dying cells during B cell culture could provide an additional TLR signal to the IgM BCR signal could not be supported (Fig. S5j, k). To further dissect the postulated IgM BCR signaling impairment in Tlr3⁻/⁻Tlr7⁻/⁻Tlr9⁻/⁻ B cells we derived B cells from single Tlr7- and Tlr9-deficient mice for analysis. We found that Tlr7-deficiency partially contributes to decreased anti-IgM reactivity which is more pronounced in Tlr3⁻/⁻Tlr7⁻/⁻Tlr9⁻/⁻ B cells (Fig. S6a).

CD23 has been implicated in the regulation of IgE in vivo[40] and is upregulated in EGT-315 Tlr3⁻/⁻Tlr7⁻/⁻Tlr9⁻/⁻ mice (Fig. S5b). In addition, a negative regulatory capacity of CD23 for BCR signaling has been shown[41]. Here, we found that stimulation with LPS leads to comparable proliferation of purified splenic B cells of C57BL/6 and Tlr3⁻/⁻Tlr7⁻/⁻Tlr9⁻/⁻ B cells (Fig. 8d). However, Tlr3⁻/⁻Tlr7⁻/⁻Tlr9⁻/⁻ B cells significantly upregulate CD23 expression in vitro to both LPS and LPS + IL-4 stimulation (Fig. 8f). This difference is Tlr4 specific because B cells stimulated with a CD40Ligand+BAFF signal (coculture with 40LB cells[37]) did not show differences in CD23 expression (Fig. S6b).

Collectively, the data emphasize that Tlr3⁻/⁻Tlr7⁻/⁻Tlr9⁻/⁻ B cells suffer from a defect in IgM BCR-mediated proliferation/survival and aberrant CD23 induction. This, together with the reduction of GC B cells, T-bet⁺ ABCs and IFN-γ production could contribute to the complete absence of GFP-specific IgM and IgG in EGT-315 Tlr3⁻/⁻Tlr7⁻/⁻Tlr9⁻/⁻ as well as ERV (MuLV) specific IgG in Tlr3⁻/⁻Tlr7⁻/⁻Tlr9⁻/⁻ mice described earlier[11].

## Discussion

Howard M. Temin's DNA provirus hypothesis of endogenous retroviruses fundamentally changed our perception of the composition of our genome[2]. Here, we describe a model that by introduction of a single unique ERV in the mouse germline enabled us to detect ERV-GFP expression already in the postnatal phase indicating that viral antigen is present early as a target for the immune system. It also implies that the tight control of ERVs in the embryo is lost after exposure of the neonate to various environmental ERV-activating signals. As expected from previous data ERV-GFP reactivation most prominently occurs in immune cells. But the strength and the particular pattern of ERV-GFP expression in some but not all crypts of the gut is noteworthy. Especially the epithelial cells of the gut border might be more sensitive and/or are strongly exposed to microbiota-driven ERV-reactivation[10]. They could also be preferred targets of secondary virus infection due to their constant cell division[42]. However, the reason why not all crypts are ERV-GFP⁺ and why Peyer's patches do not reactivate ERV-GFP remains to be determined. Nevertheless, this suggests that the hypothesis of exposure of ERV-activating substances from the microbiota as the cause of ERV viremia might be too simplistic[10].

Backcrosses of EGT-315 mice over a number of generations revealed that the expression of ERV-GFP while declining through the generations was still sufficient to induce anti-GFP antibody responses. The probable cause of the decreased ERV-GFP expression could not be found in increase of DNA-methylation of the ERV-GFP, but rather was caused by a reduction in the 129/Sv-encoded ERV-permissive modifier genes. Positional cloning could be a way to identify the underlying genetic differences between C57BL/6 and 129/Sv mice to identify unknown host factors that are involved in ERV control.

The study of MuLVs laid the foundation for how exogenous retroviruses interact with the immune system[43,44]. Here we resolve the dichotomy of ERVs being both part of the genome (=self) and infectious virus (=foreign) by showing the underlying mechanisms of a self-specific antibody response against the host's own ERVs[13].

By examining the anti-GFP response we unmask that the B cell response against ERVs is only directed against proteins which are part of the virus as in EGT-315 B6 mice where GFP is fused to the env-protein making it an integral part of the virus envelope. The absence of anti-GFP IgG in IRES-GFP containing ERV-GFP mice refutes the idea that ERV reactivation per se could break tolerance to GFP. Also, various GFP-reporter mice do not mount anti-GFP antibody responses. Taken together, this points to the virus-particle as the trigger and target of the immune response[9].

Before the virus buds from the cells Env-GFP is present on the cell surface of ERV-GFP expressing cells. Historically, it was suggested that high-affinity recognition of membrane self-antigens leads to complete tolerance of self-reactive B cells[45]. But in our model it does not lead to tolerance but rather to antibodies against Env-GFP. This matches

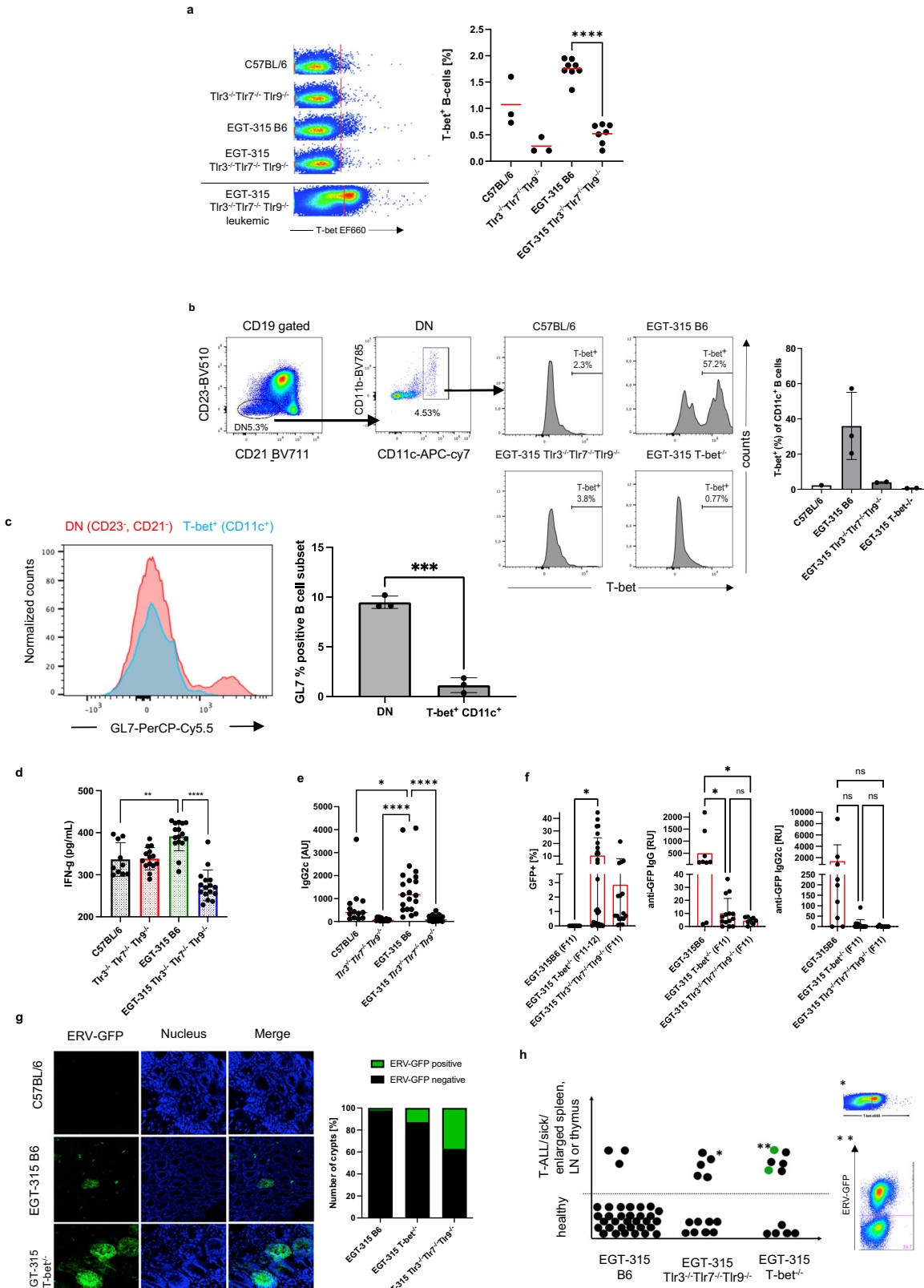

recent ideas that mild autoreactivity of the BCR is required to ensure proper B cell development[46] indicating that the mature B cell repertoire is capable of recognizing various self-structures. Further, it has been suggested that antigen-valency plays a crucial role in the activation of autoreactive B cells[47,48] Nevertheless, valency alone is not sufficient to establish a longlasting autoantibody response possibly due to the lack of costimulatory signals. Interestingly, our model provides a polyvalent autoantigen (Env-GFP) together with TLR7 agonist (viral RNA) as a costimulatory signal, which is most likely triggering a long-lasting autoantibody response accompanied by IC kidney deposition. This also suggests that ERVs are the cause for spontaneous GC-induction, similar to the previously described Tlr7-dependent function

**Fig. 5 | ERV-GFP recognition results in upregulation of IFN-γ, IgG2c, increases T-bet⁺ ABCs and is dependent on T-bet. a** T-bet positive B cells identified by intracellular flow cytometry. Left, examples of dot plot analysis of the four genotypes, red line demarcates T-bet negative vs positive cells. Lower example, cells of a leukemic EGT-315 Tlr3⁻/⁻Tlr7⁻/⁻Tlr9⁻/⁻ mouse. Right, summary of % T-bet⁺ cells of individual mice: C57BL/6 (mean = 1.07, SD = 0.463, n = 3), Tlr3⁻/⁻Tlr7⁻/⁻Tlr9⁻/⁻ (mean = 0.29, SD = 0.15, n = 3), EGT-315 B6 (mean = 1.75, SD = 0.189, n = 8), EGT-315 Tlr3⁻/⁻Tlr7⁻/⁻Tlr9⁻/⁻ (mean = 0.52, SD = 0.19, n = 7). EGT-315 B6 vs. EGT-315 Tlr3⁻/⁻Tlr7⁻/⁻Tlr9⁻/⁻ ****P < 0.0001. Ordinary one way ANOVA multiple comparisons test. Summary of 5 experiments. **b** ABCs in EGT-315 B6 mice. Gating strategy of CD19⁺ splenic B cells from C57BL/6 (2.3, n = 1), EGT-315 B6 (mean = 35.97, SD = 19.1, n = 3), EGT-315 TLR3⁻/⁻TLR7⁻/⁻TLR9⁻/⁻ (mean = 4.0, SD = 0.33, n = 2) and EGT-315 T-bet⁻/⁻ mice (mean = 0.67, SD = 0.15, n = 2). DN defines CD23⁻ CD21⁻ B cells. Box in DN panel defines both CD11c⁺ CD11b⁺ and CD11c⁺ CD11b⁻ B cells analyzed for T-bet expression. Right panel, summary of mice tested for % T-bet positive CD11c⁺ cells. **c** Overlay of histogram for GL7 of DN (red) and T-bet⁺ CD11c⁺ ABC population (blue) of EGT-315 B6 mice (n = 3). GL7 positive B cells are present in the DN B cell population but not in the ABC population in spleen of individual EGT-315 B6 mice. DN (mean = 9.48 SD = 0.627, n = 3), T-bet⁺ CD11c⁺ ABC (mean = 1.14 SD = 0.752, n = 3), ***P = 0.0002. Unpaired Two-tailed t-test with Welch´s correction. **d** Serum IFN-γ levels in pg/ml measured by cytometric bead array for mouse inflammatory cytokines in individual sera of C57BL/6 (black bar, mean = 337, SD = 40, n = 10), Tlr3⁻/⁻Tlr7⁻/⁻Tlr9⁻/⁻ (red bar, mean = 338, SD = 26, n = 14), EGT-315 B6 (green bar, mean = 392, SD = 34, n = 16), EGT-315 Tlr3⁻/⁻Tlr7⁻/⁻Tlr9⁻/⁻ (blue bar, mean = 275, SD = 37, n = 16). For statistics a Two-tailed Mann–Whitney U-test was applied. EGT-315 B6 vs. C57BL/6 **P = 0.0029, EGT-315 B6 vs. EGT-315 Tlr3⁻/⁻Tlr7⁻/⁻Tlr9⁻/⁻ ****P < 0.0001. Experiment is a summary of two experiments. **e** Serum IgG2c levels measured by IgG2c-specific ELISA in arbitrary units (AU) in C57BL/6 (mean = 683, SD = 918, n = 13), Tlr3⁻/⁻Tlr7⁻/⁻Tlr9⁻/⁻ (mean = 83, SD = 62, n = 14), EGT-315 B6 (mean = 1442, SD = 1088, n = 21). EGT-315 Tlr3⁻/⁻Tlr7⁻/⁻Tlr9⁻/⁻ (mean = 121, SD = 120, n = 23) mice. C57BL/6 vs EGT-315 B6 *P = 0.0186; Tlr3⁻/⁻Tlr7⁻/⁻Tlr9⁻/⁻ vs EGT-315 B6 ****P < 0.0001; EGT-315 B6 vs EGT-315

Tlr3⁻/⁻Tlr7⁻/⁻Tlr9⁻/⁻ ****P < 0.0001. Ordinary one-way ANOVA Tukey´s multiple comparisons test. Summary of 3 experiments. **f** ERV-GFP expression and anti-GFP antibody response in EGT-315 T-bet⁻/⁻ mice. Flow cytometry of generation matched (F11-F12) EGT-315 B6 mice (mean = 0, SD = 0, n = 9) EGT-315 T-bet⁻/⁻ (mean = 10.5, SD = 14.1, n = 26) and EGT-315 Tlr3⁻/⁻Tlr7⁻/⁻Tlr9⁻/⁻ mice (mean = 2.9, SD = 5.2, n = 17). % GFP positive cells in PBMC of 3-week old mice (left panel). EGT-315 B6 vs EGT-315 T-bet⁻/⁻ *P = 0.0327. 10 out of 26 mice (38.5%) of the EGT-315 T-bet⁻/⁻ mice display very high ERV-GFP expression (>10%). Red bars are mean values. Statistics with Ordinary one-way ANOVA Tukey´s multiple comparisons test. Summary of 3 experiments. Anti-GFP IgG (middle) EGT-315 B6 mice (mean = 504, SD = 773, n = 9) EGT-315 T-bet⁻/⁻ (mean = 10.2, SD = 11.3, n = 14) and EGT-315 Tlr3⁻/⁻Tlr7⁻/⁻Tlr9⁻/⁻ mice (mean = 4.5, SD = 2.5, n = 8). EGT-315 B6 vs EGT-315 T-bet⁻/⁻ *P = 0.0242; EGT-315 B6 vs EGT-315 Tlr3⁻/⁻Tlr7⁻/⁻Tlr9⁻/⁻ *P = 0.0485; EGT-315 T-bet⁻/⁻ vs EGT-315 Tlr3⁻/⁻Tlr7⁻/⁻Tlr9⁻/⁻ ˢⁿP = 0.9995; Statistics with Tukey´s multiple comparisons test. Anti-GFP IgG2c (right panel) response of 2–3 month old mice measured by ELISA. EGT-315 B6 mice (mean = 1374, SD = 2873, n = 9) EGT-315 T-bet⁻/⁻ (mean = 9.3, SD = 24.3, n = 14) and EGT-315 Tlr3⁻/⁻Tlr7⁻/⁻Tlr9⁻/⁻ mice (mean = 1, SD = 3, n = 8). All ˢⁿP-values > 0.1. Summary of 2 experiments. **g** Confocal microphotograph from small intestine. GFP signal indicates ERV-GFP expression. Intestines of 2.5-month old mice representative for C57BL/6, EGT-315 B6 and EGT-315 T-bet⁻/⁻ mice. Right, summary of ERV-GFP positive crypts (green) compared to negative crypts (black) EGT-315 B6 mice (2.2% GFP⁺, total crypts analyzed 792 of n = 21), EGT-315 T-bet⁻/⁻ (12.8% GFP⁺, total crypts analyzed 601 of n = 9) and EGT-315 Tlr3⁻/⁻Tlr7⁻/⁻Tlr9⁻/⁻ mice (37.2% GFP⁺, total crypts analyzed 487 of n = 16). Summary of 4 experiments. **h** EGT-315 T-bet⁻/⁻ mice succumb to T-ALL. Summary of mice analysed: EGT-315 B6 (3/34, 8.8%). EGT-315 Tlr3⁻/⁻Tlr7⁻/⁻Tlr9⁻/⁻ (5/13, 38%). EGT-315 T-bet⁻/⁻ (6/11, 55%). The mice examined were 7 month or older. Each dot represents a single mouse. *T-ALL cells from an EGT-315 Tlr3⁻/⁻Tlr7⁻/⁻Tlr9⁻/⁻ mouse show increased T-bet expression. **Two out of six lymphomas of EGT-315 T-bet⁻/⁻ mice were ERV-GFP positive (green dots). Source data are provided as a Source Data file.

---

in exogenous Friend virus infection[49]. In autoimmune models Tlr7-dependent spontaneous GC formation has been observed[32,50]. It is supported by data of RNA-associated autoantigen specific antibody response triggered by joint BCR and TLR7 signals[51,52].

These similarities between anti-ERV response and B cell auto-immunity extend to IFN-γ dependence[53,54] and consequently to an essential role of ABCs, T-bet⁺ memory B cells in both responses. The increased IFN-γ expression in ERV-GFP Tlr-competent mice is surprising because, although it is an important cytokine for priming anti-(retro)viral immune responses and T-bet⁺ memory B cells[55], it has not been shown previously to be induced by ERVs. Our data suggest an immune-axis that starts with (A) the expression of ERV in bone marrow or gut which leads to (B) TLR sensing of viral RNA/DNA by cells of the innate immune system that drives (C) IFN-γ production. This jointly induces (D) the expression of T-bet in B cells which in turn upregulates activation-induced cytidine deaminase (AID) and ultimately leads to selection and expansion of B cells with class switched high-affinity (E) ERV-specific BCR. Hence, this ERV specific B cell response is able to neutralize virus and thereby impedes reinfection of proliferating cells. ERV-specific antibodies work in concert with epigenetic control to ultimately protect against insertional mutagenesis by uncontrolled ERVs. Notwithstanding, by whole-genome sequencing we were able to confirm the single germline ERV-GFP insertion site in Chr.19, but a single additional ERV-GFP insertion in spleen cells could be located in Chr.13. We speculate that this is a somatic ERV-GFP insertion which occurred in spleen cells of the EGT-315 B6 mouse and is in line with continous detection of ERV-GFP⁺ antigens in spleen. The elimination of ERV-expressing cells by the immune system and subsequent replacement of these cells, e.g. immune- and gut epithelia cells may avoid crossing the threshold of inflammation and explain why overt auto-immune pathology does not manifest in ERV-GFP mice.

With regard to the mechanism how more T-bet⁺ ABCs are generated in ERV-GFP mice but not in TLR-compromised mice our experiments reveal a B cell specific defect of Tlr3⁻/⁻Tlr7⁻/⁻Tlr9⁻/⁻ (NAS-Tlrs)

mice. It was shown previously that CD23 can negatively regulate BCR signaling. CD23⁻/⁻ B cells showed increased phosphorylation of Erk, Akt and Btk compared to CD23⁺/⁺ B cells upon BCR crosslinking in vitro[41]. Here, we show that B cells from Tlr3⁻/⁻Tlr7⁻/⁻Tlr9⁻/⁻ mice react normally to CD40L stimulation. Interestingly, Tlr3⁻/⁻Tlr7⁻/⁻Tlr9⁻/⁻ B cells show increased CD23 expression upon LPS (Tlr4-ligand) stimulation. Elevated CD23 expression levels could contribute to the impaired pro-liferation/survival of B cells upon anti-IgM crosslinking in the absence of NAS Tlrs observed earlier. Although, IgM-BCR- and TLR-signaling have been extensively studied[56,57], no coherent mechanistic explanation exists how the lack of the NAS-TLRs affects IgM-BCR signaling and whether an effect beyond impaired ERV control exists. It is conceivable that NAS-TLRs may control IgM homeostasis and contribute to rapid IgM responses during infections. Future studies should address the questions whether lack of NAS-TLRs affects biochemical properties (polyreactivity) as well as functions of natural IgM and whether this is connected to T-bet⁺ B cells[17]. A ligand-independent tonic signal through NAS-TLRs might influence the IgM-BCR signaling in a way that it reaches a certain threshold required for differentiation into IgM effector cells. Tlr7 single-deficient mice show a slight reduction of proliferation/survival in vitro making Tlr7 a candidate for having enhancing effects on BCR-signaling. Interestingly, and consistent with our data on decreased T-bet expression and B cell proliferation in NAS-Tlr deficient mice, a recently described lupus-inducing TLR7 gain-of-function mutation shows both increased survival/proliferation and T-bet induction in B cells after IgM crosslinking[58]. Complementary, during the SARS-CoV-2 pandemic humans with TLR7-deficiency suffering from severe COVID-19 were identified[59]. At the moment it is still unclear, whether this is a result of the reduced IFN-γ response due to impaired RNA-virus sensing in these patients or a consequence of a TLR7-related defect that affects the B cell response to SARS-CoV-2[60].

Finally, our model supports the hypothesis that ERVs still shape the human immune response[35]. Firstly, the de novo introduction of the ERV-GFP increases anti-DNA antibody titers in vivo. ERV-expression,

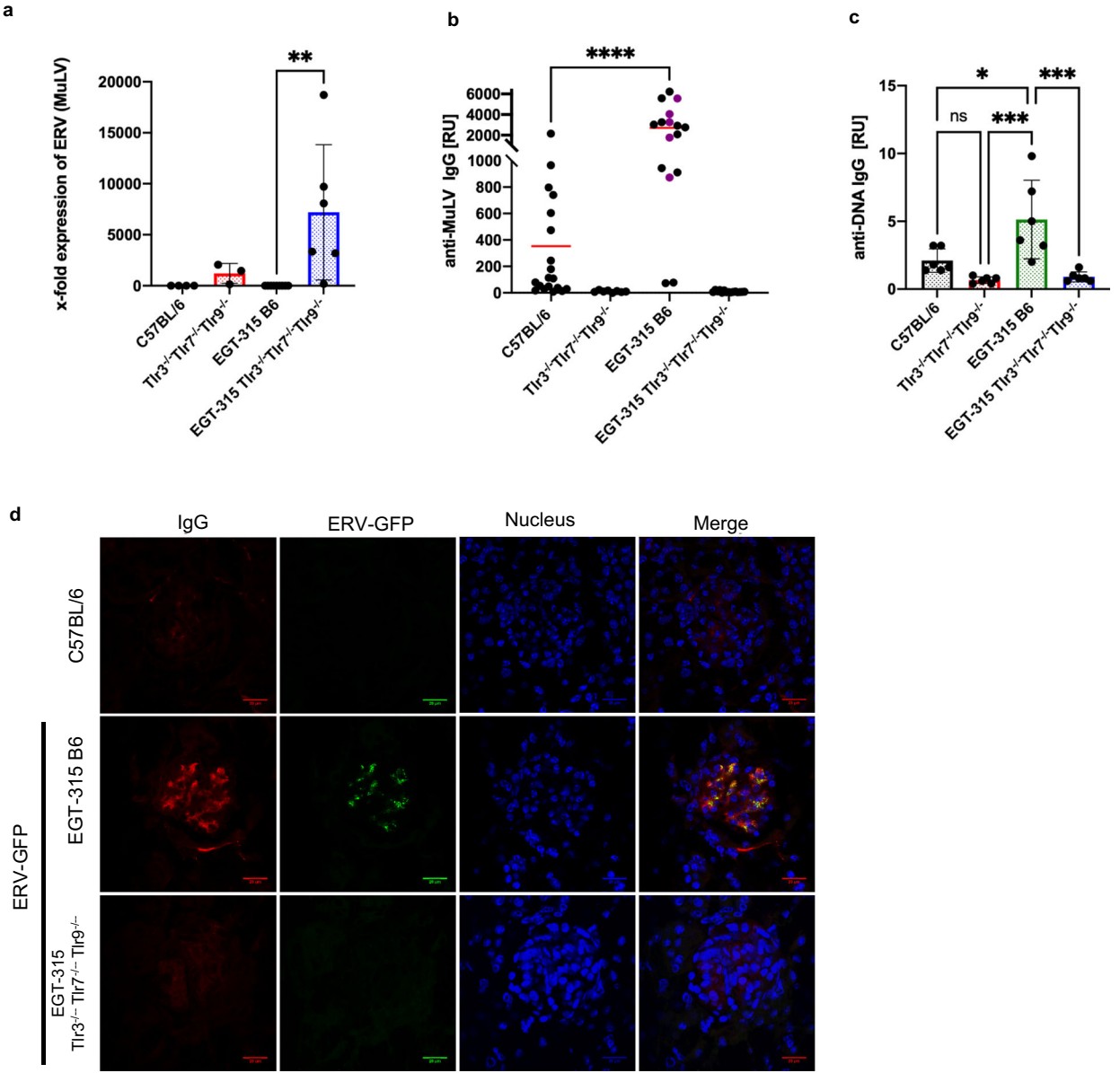

**Fig. 6 | ERV-GFP enhances natural ERV (MuLV of C57BL/6) expression, anti-MuLV antibody production, anti-DNA IgG production and immune complex deposition in kidneys. a** MuLV specific mRNA expression analyzed by Q-PCR of purified splenic B cells from individual mice. Actin was used to normalize expression. ERV (MuLV): C57BL/6 (black bar, mean = 0.04, SD = 0.03, $n = 4$), $Tlr3^{-/-}Tlr7^{-/-}Tlr9^{-/-}$ (red bar, mean = 1216, SD = 800, $n = 4$), EGT-315 B6 (green bar, mean = 0.89, SD = 0.23, $n = 9$), EGT-315 $Tlr3^{-/-}Tlr7^{-/-}Tlr9^{-/-}$ (blue bar, mean = 7202, SD = 6050, $n = 6$), Summary of 6 experiments; **$P = 0.0055$ by unpaired Two-tailed t test. **b** Anti-MuLV IgG production is increased in EGT-315 B6 mice. MuLV-specific ELISA of plasma from C57BL/6 (mean = 252.7, SD = 315.4, $n = 19$), $Tlr3^{-/-}Tlr7^{-/-}Tlr9^{-/-}$ (mean = 11.9, SD = 7.3, $n = 8$), EGT-315 B6 (mean = 2711, SD = 1936, $n = 16$ including 5 pools purple dots), EGT-315 $Tlr3^{-/-}Tlr7^{-/-}Tlr9^{-/-}$ (mean = 9.1, SD = 6.7, $n = 15$

including 5 pools), red lines depict mean value. Summary of 3 experiments; C57BL/6 vs EGT-315 B6 ****$P < 0.0001$. Statistics with Ordinary one-way ANOVA Tukey´s multiple comparisons test. **c** Anti-DNA IgG production is increased in EGT-315 B6 mice. DNA (ss and ds) specific ELISA of plasma from C57BL/6 (black bar, mean = 2.1, SD = 0.86, $n = 6$), $Tlr3^{-/-}Tlr7^{-/-}Tlr9^{-/-}$ (red bar, mean = 0.67, SD = 0.24, $n = 6$), EGT-315 B6 (green bar, mean = 5.1, SD = 2.9, n = 6), EGT-315 $Tlr3^{-/-}Tlr7^{-/-}Tlr9^{-/-}$ (blue bar, mean = 0.9, SD = 0.37, $n = 6$). **$P = 0.0055$. Statistics with Ordinary one-way ANOVA Tukey´s multiple comparisons test. **d** Confocal microphotography of immune complex deposition containing IgG and ERV-GFP in kidney glomeruli representative of EGT-315 B6 ($n = 7$) but not in C57BL/6 ($n = 4$) and EGT-315 $Tlr3^{-/-}Tlr7^{-/-}Tlr9^{-/-}$ mice ($n = 2$). Anti-IgG Alexa Fluor 633, DAPI (nucleus) and ERV-GFP are shown. Scale bar 20 µm. Source data are provided as a Source Data file.

-reinfection and -replication could provide signals to anti-DNA BCR expressing B cells and therefore lead to the observed increase in anti-DNA antibody titers in vivo. Secondly, ERV-GFP expression induces enhanced secretion of IFN-γ into the serum. We assume that this steady state increase is a direct effect of the continous reactivation of ERV-GFPs monitored by NAS-TLRs. Thirdly, the presence of ERV-GFP only in conjunction with lack of NAS-TLRs increases IgE levels. This switch from ERV-GFP specific IgG2c to unspecific IgE in the NAS-TLR-deficient mice might be due to the well established antagonistic nature of IFN-γ

(Th-1) and IL-4 (Th-2)[61]. Reduced IFN-γ production could lead to baseline IL-4 levels that cause a default Th-2 T cell response and switch to IgE. Consensus exists that Tlr9-engagement upregulates T-bet and IgG2a/c in B cells, however whether its deletion increases IgE is not clear[62,63]. Intriguingly, an alternative explanation could be that increased ERV expression in the absence of TLRs might be sensed by other retrovirus- or nucleic-acid sensors[64] which directly induce IgE expression in vivo without the involvement of IL-4. This possibility is supported by data which showed IgE induction in the absence of IL-4

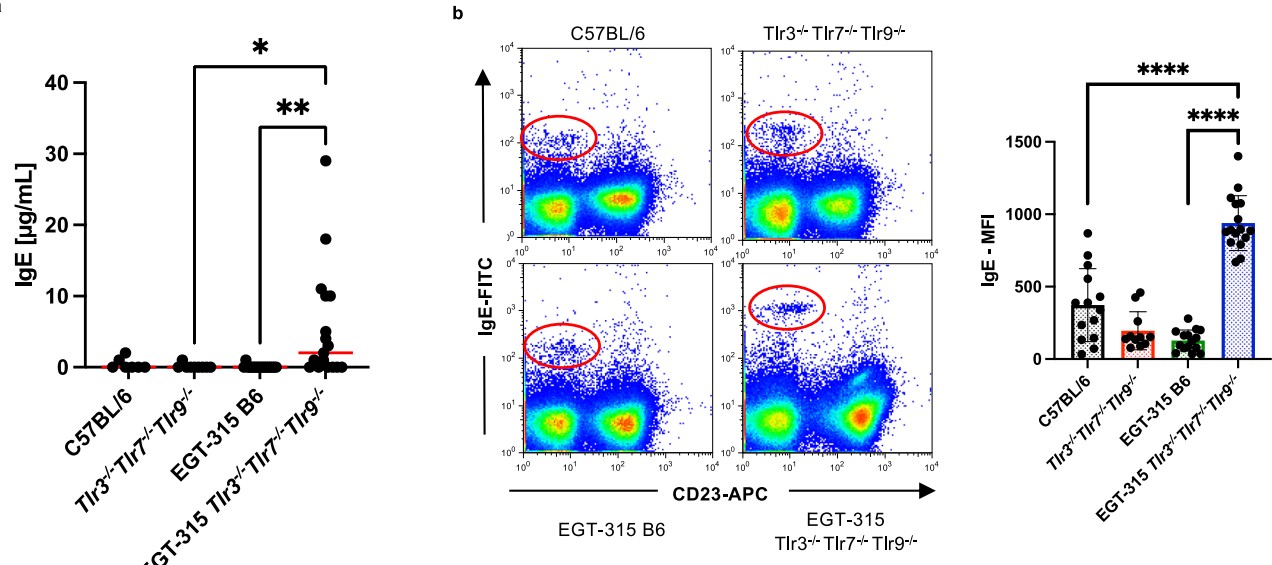

**Fig. 7 | Presence of ERV-GFP and absence of nucleic-acid Tlr-sensing results in systemic IgE overproduction. a** Serum IgE levels measured by IgE-specific ELISA from individual mice: C57BL/6 (mean = 0.43, SD = 0.79, $n$ = 7), Tlr3$^{-/-}$Tlr7$^{-/-}$Tlr9$^{-/-}$ (mean = 0.11, SD = 0.33, $n$ = 9), EGT-315 B6 (mean = 0.06, SD = 0.24, $n$ = 18) EGT-315 Tlr3$^{-/-}$Tlr7$^{-/-}$Tlr9$^{-/-}$ (mean = 5.5, SD = 8.0, $n$ = 17). Red lines depict mean value. Tlr3$^{-/-}$Tlr7$^{-/-}$Tlr9$^{-/-}$ vs EGT-315 Tlr3$^{-/-}$Tlr7$^{-/-}$Tlr9$^{-/-}$ *$P$ = 0.035. EGT-315 B6 vs EGT-315 Tlr3$^{-/-}$Tlr7$^{-/-}$Tlr9$^{-/-}$ **$P$ = 0.0061. Ordinary one-way ANOVA Tukey´s multiple comparisons test. Summary of 3 experiments. **b** Basophil-bound IgE levels measured by flow cytometry of spleen cells from individual mice: C57BL/6 (mean = 373, SD = 252; $n$ = 14), Tlr3$^{-/-}$Tlr7$^{-/-}$Tlr9$^{-/-}$ (mean = 196, SD = 131, $n$ = 11), EGT-315 B6 (mean = 128, SD = 73, $n$ = 15), EGT-315 Tlr3$^{-/-}$Tlr7$^{-/-}$Tlr9$^{-/-}$ (mean = 939, SD = 190, $n$ = 16). Left exemplary flow cytometry panel, and right panel statistical evaluation of MFI of the IgE$^+$CD23$^-$ basophil population (red circle). C57BL/6 vs EGT-315 Tlr3$^{-/-}$Tlr7$^{-/-}$Tlr9$^{-/-}$ ****$P$ < 0.0001. EGT-315 B6 vs EGT-315 Tlr3$^{-/-}$Tlr7$^{-/-}$Tlr9$^{-/-}$ ****$P$ < 0.0001. Statistics with Ordinary one-way ANOVA Tukey´s multiple comparisons test. Summary of 12 experiments. Source data are provided as a Source Data file.

or CD40 by LP-BM5 MuLV, a defective retrovirus which causes anergy and lymphoproliferation[65,66].

Our model provides the insight that the mammalian immune system constantly surveys endogenous retrovirus reactivation from within the genome. It overcomes self-tolerance by engaging innate nucleic-acid sensors and eventually activates an ERV-specific T-bet$^+$ B cell mediated response. It is tempting to speculate that this sequence of mechanisms originally evolved to control ERVs is also involved in human autoimmunity.

## Methods

### Ethics approval
All mice were kept in specific pathogen-free isolator cages and handled according to the German animal experimentation law. All animal experiments and procedures were in accordance with institutional guidelines and were approved under the permit references V54-19c 20 15 h 01 MR 20/8 (Nr 84/2014, Nr G63/2019 and Nr G80/2021) by the Regierungspräsidium Gießen.

### Mice and cell lines
Mice were kept under Specific pathogen free (SPF) conditions in IVC cages with a 12 h/12 h light dark cycle, a humidity of 55% +/- 5% at a room temperature of 21 +/− 1 °C. The following mouse strains were used in this study: C57BL/6 J from Charles River (Germany); hA3 transgenic mice[26], which contain the human Apobec3 gene locus, were crossed to generate EGT-315 Tlr3$^{-/-}$Tlr7$^{-/-}$Tlr9$^{-/-}$ hA3 mice; Tlr7[67]-and Tlr9[68]-deficient mice were used to isolate primary B cells for in vitro activation. Tlr3$^{-/-}$Tlr7$^{-/-}$Tlr9$^{-/-}$ triple deficient[11] mice; and T-bet deficient C57BL/6 mice (Tbx21 deficient[69]). For in vitro infections with ERV-GFP we used the B cell lymphoma WEHI-231 (ATCC CRL-1702), NIH-3T3 cell line (ATCC CRL-1658) and primary mouse embryonic fibroblasts (MEF) isolated from C57BL/6 mice. HEK-Blue hTLR7 Cells (InvivoGen) and HEK mTLR9 cells were used as reporter cells for Tlr7 and Tlr9 ligands

respectively[70]. For the indirect immunofluorescence assay HEp-2 cells (ATCC CCL-23) were used. For in vitro B cell activation we used 40LB cells[37].

### Generation of ERV-GFP strain EGT-315 (Endogenous retrovirus-GFP-Tagged)
EGT-315 were generated using the embryonic stem cell V6.5f1(C57BL/6 x 129/Sv)[71] a gift from Thomas Wunderlich, Max Planck Institute for Metabolism Research, Cologne, Germany. ES cells were cultured on mouse embryonic fibroblasts (from C57BL/6) and infected using pMOV-GFP[16] derived Mo-MuLV, which was isolated from the supernatant of retrovirus producing NIH-3T3 cells. For ERV-GFP analysis an aliquot of the infected ES cells was enriched by 3 passages of feeder free culture on cell culture dishes coated with 0.5% gelatin in PBS. Infected MEFs were enriched by differential passaging of the MEFs followed by flow cytometric counterstaining with CD90.2 antibody. C57BL/6 blastocysts were injected with pMOV-GFP PCR positive pools of ES cells and chimeric founder mice were backcrossed to C57BL/6 J (8−11 generations) or Tlr3$^{-/-}$Tlr7$^{-/-}$Tlr9$^{-/-}$ triple deficient mice on C57BL/6 background (11 generations).

### Flow cytometry-, immunohistochemically- and in situ GFP expression analysis
Flowcytometry analysis was done with single cell suspensions of organs of EGT-315 B6, EGT-315 Tlr3$^{-/-}$Tlr7$^{-/-}$Tlr9$^{-/-}$ mice, wild type littermates or C57BL/6 and Tlr3$^{-/-}$Tlr7$^{-/-}$Tlr9$^{-/-}$ mice. They were analyzed with a FacsCalibur Cytometer (Becton Dickinson). To block unspecific Ab binding on Fc receptors ChromPure rat IgG whole molecule (1:75, Jackson Immuno Research) was incubated for 7 min at 4 °C. For staining the antibodies listed in the Supplementary Table 1 were used at 1:300 dilution. For counterstaining of biotinilated Abs Streptavidin-PerCP (BioLegend) was used. After incubation for 30 min at 4 °C cells were washed and measured. For histological analysis spleens, kidneys,

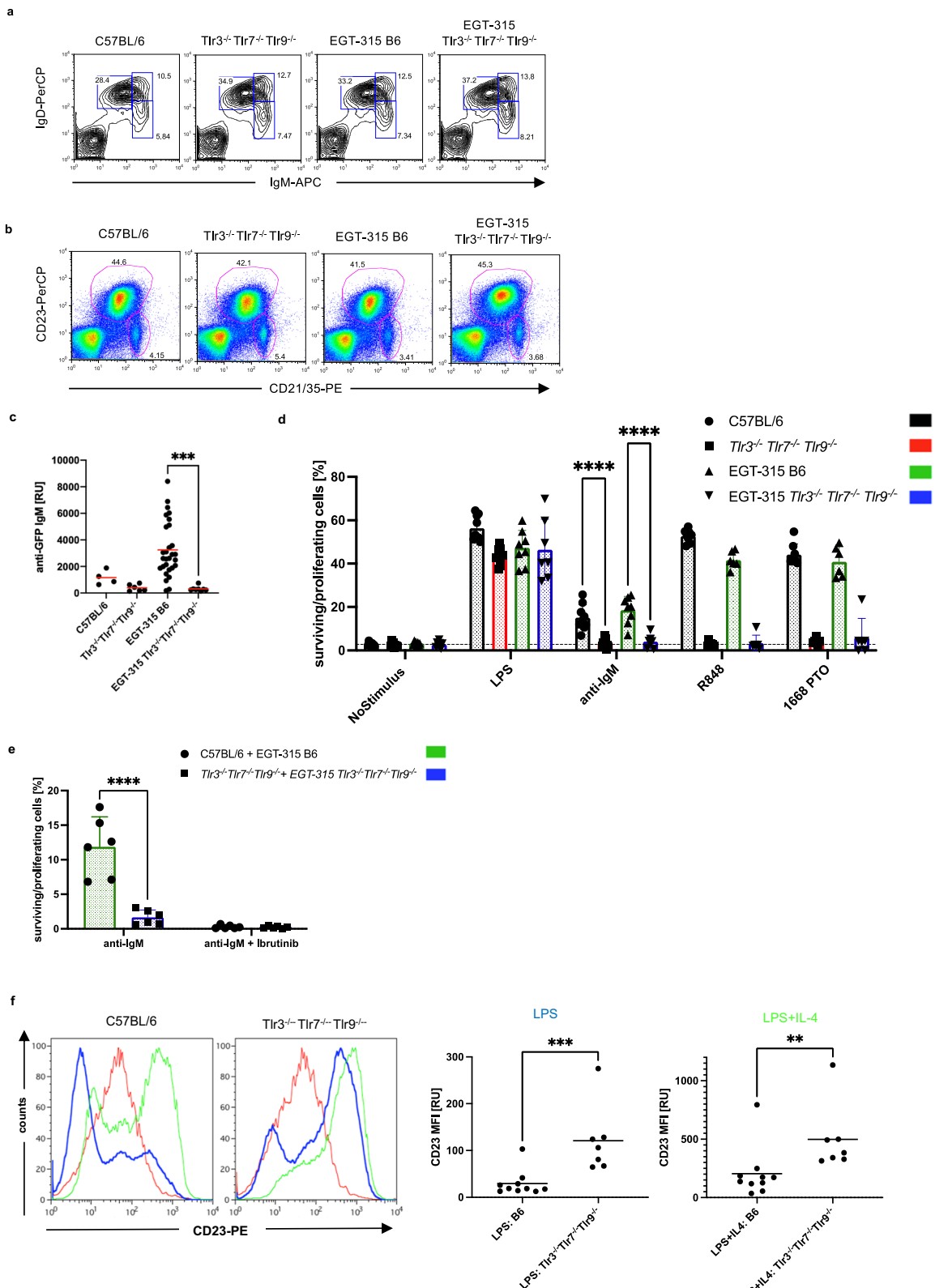

small and large intestines from different mice were shock frozen in cryo-embedding medium and 14–18 micrometer cryosections were prepared. The cryosections were fixed on slides with acetone for 5 min at −20 °C and air-dried for 15 min. To block unspecific Ab binding on Fc receptors ChromPure rat IgG whole molecule (1:100, Jackson Immuno Research) was used. After incubation for 30 min sections were washed in PBS with 1 % BSA to remove excess blocking Ab. Spleen sections

were stained with anti-CD23 PE (1:50, clone B3B4; BD Biosciences), anti-IgG (H + L) Alexa Fluor 633 antibody (1:100, Thermo Fisher Scientific), anti-CD21/CD35 PE (1:25, clone 7E9; Miltenyi Biotec), anti-CD23 APC (1:50, clone B3B4, Thermo Fisher Scientific), anti-F4/80 PE (1:100, clone BM8; Thermo Fisher Scientific), and anti-CD4 APC (1:100, clone RM4-5; BD Biosciences), peanut agglutinin (PNA)-Alexa Fluor 647 (1:50, Molecular Probes). Kidney sections were stained with anti-IgG (H + L)

**Fig. 8 | Combined deficiency of Tlr3 Tlr7 Tlr9 in B cells impairs IgM-BCR induced B cell survival/proliferation but upregulates CD23 expression induced by Tlr4 in vitro. a** Flow cytometry plots of IgM vs IgD staining of splenic B cells representative of C57BL/6 ($n = 7$), Tlr3$^{-/-}$Tlr7$^{-/-}$Tlr9$^{-/-}$ ($n = 7$), EGT-315 B6 ($n = 6$), EGT-315 Tlr3$^{-/-}$Tlr7$^{-/-}$Tlr9$^{-/-}$ ($n = 6$). Squares: mature B cells (left), transitional B cells (right upper), immature B cells (right lower). Summary of 9 experiments. **b** Follicular (gate upper middle) and marginal zone (gate lower right) B cell distribution in Tlr deficient mice. Exemplary flowcytometry of spleen cells from C57BL/6 ($n = 3$), Tlr3$^{-/-}$Tlr7$^{-/-}$Tlr9$^{-/-}$ ($n = 2$), EGT-315 B6 ($n = 6$), EGT-315 Tlr3$^{-/-}$Tlr7$^{-/-}$Tlr9$^{-/-}$ ($n = 5$) using the B cell markers CD23 and CD21. Summary of 3 experiments. **c** GFP-specific IgM measured by GFP specific ELISA from individual mice: C57BL/6 (mean = 1160, SD = 476, $n = 4$), Tlr3$^{-/-}$Tlr7$^{-/-}$Tlr9$^{-/-}$ (mean = 392, SD = 229, $n = 6$), EGT-315 B6 (mean = 3249, SD = 2037, $n = 29$), EGT-315 Tlr3$^{-/-}$Tlr7$^{-/-}$Tlr9$^{-/-}$ (mean = 312, SD = 171, $n = 8$). Red lines depict mean value. EGT-315 B6 vs. EGT-315 Tlr3$^{-/-}$Tlr7$^{-/-}$Tlr9$^{-/-}$ ***$P = 0.0004$. Ordinary one-way ANOVA Tukey's multiple comparisons test. **d** Survival/proliferation of purified B cells isolated from individual mice 72 h after in vitro stimulation (no stimulus, LPS = Tlr4 ligand, anti-IgM = BCR ligand, R848 = Tlr7 ligand, 1668 PTO = Tlr9 ligand) measured by flow cytometry using propidium iodide to distinguish live vs dead cells. C57BL/6 (black bar, for anti-IgM: mean = 15.1, SD = 5.9, $n = 9$), Tlr3$^{-/-}$Tlr7$^{-/-}$Tlr9$^{-/-}$ (red bar, for anti-IgM: mean = 3.2, SD = 2.0, $n = 9$), EGT-315 B6 (green bar, for anti-IgM: mean = 18.6, SD = 6.3, $n = 8$), EGT-315 Tlr3$^{-/-}$Tlr7$^{-/-}$Tlr9$^{-/-}$ (blue bar, for anti-IgM: mean = 4.0, SD = 3.0, $n = 7$). Summary of 12 experiments, 2 way ANOVA, Sidak´s multiple comparisons test ****$P < 0.0001$. **e** Role of Btk in survival/proliferation of purified B cells 72 h after in vitro stimulation. Anti-IgM was used with and without the addition of the Btk-inhibitor Ibrutinib. Cells from individual mice were tested. C57BL/6 and EGT-315 B6 combined (green bar, for anti-IgM: mean = 11.9, SD = 4.3, $n = 6$), Tlr3$^{-/-}$Tlr7$^{-/-}$Tlr9$^{-/-}$ and EGT-315 Tlr3$^{-/-}$Tlr7$^{-/-}$Tlr9$^{-/-}$ B6 combined (blue bar, for anti-IgM: mean = 1.7, SD = 1.1, $n = 6$). Summary of 5 experiments, 2-way ANOVA, Sidak´s multiple comparisons test, ****$P < 0.0001$. **f** Tlr3$^{-/-}$Tlr7$^{-/-}$Tlr9$^{-/-}$ B cells induced by LPS (Tlr4-ligand) upregulate CD23. Upper panels, LPS-mediated in vitro induction of CD23 is higher in Tlr3$^{-/-}$Tlr7$^{-/-}$Tlr9$^{-/-}$ B cells compared to wild type B cells. Flowcytometry examples of stimulated B cells from C57BL/6 and Tlr3$^{-/-}$Tlr7$^{-/-}$Tlr9$^{-/-}$ mice for 72 h with medium alone (red), LPS (10 µg/ml, blue) and LPS + IL-4 (LPS 10 µg/ml+ IL-4 250 U/ml, green). Right panels, Summary of MFI of CD23 from three independent experiments. Mann-Whitney test Two-tailed pairwise comparison ***$P = 0.0007$ and **$P = 0.0031$. B6 ($n = 10$), Tlr3$^{-/-}$Tlr7$^{-/-}$Tlr9$^{-/-}$ ($n = 7$). Source data are provided as a Source Data file.

Alexa Fluor 633 antibody (1:100, Thermo Fisher Scientific). Small intestine sections were stained with anti-CD4 PE (1:100, clone GK1.5; Thermo Fisher Scientific) and anti-CD45R/B220 APC antibody (1:100, clone RA3-6B2; BD, Biosciences). For all organ sections DAPI (1:5000, Thermo Fisher Scientific) was used for nuclear counterstaining. After 1 h incubation three wash steps were done in PBS with 1 % BSA. Finally, the samples were washed with PBS without BSA and mounted with Mowiol. Images were captured using a confocal laser scanning microscope (LEICA TCS SP5 II) and analyzed with Fiji software. The signal intensity of the stainings, but not of ERV-GFP, was slightly enhanced for better visualization and was carried out consistently between control and test sections. For Ca$^{2+}$-mobilization splenic B cells were loaded with Fluo-4 (2.5 µM for 20 min at room temperature) and then analyzed by flow cytometry. After establishment of baseline Ca$^{2+}$ concentration cells were stimulated by anti-IgM (25 µg/ml in HBSS + 2 mM Ca$^{2+}$; STAR86, BioRad) for additional 4 min. Data was analyzed by FlowJo kinetic mode.

The in-situ GFP ex vivo whole-organ imaging was used to compare the ERV-GFP signal between multiple types of lymphoid and non-lymphoid tissues. Therefore, the spleen, gut, thymus, lymph nodes, brain, liver and kidney were isolated. The organs were placed on a petri dish and analyzed with an In Vivo Xtreme imaging system (Bruker Bio Spin, Hanau, Germany). Fluorescence images were captured using a 480 nm excitation filter, 535 nm emission filter and 1 s. exposure time. Image acquisition as well as analysis was performed using the Bruker MI SE software.

## ELISA, western blot and ELISPOT

Determination of anti-GFP IgG titers in Fig. 3a, b. For ELISA 96 well-plates (Maxisorp, Nunc) were coated with 50 ng GFP (Abcam) in 50 ml PBS per well. Plasma from 2- to 4-month old EGT-315 B6 (ERV-GFP high $n = 5$; ERV-GFP low $n = 11$; ERV-GFP negative $n = 14$); EGT-315 Tlr7-deficient mice (genotype: EGT-315-Tlr3$^{+/-}$Tlr7$^{-/-}$Tlr9$^{+/-}$, $n = 6$), EGT-315 Tlr3$^{-/-}$Tlr7$^{-/-}$Tlr9$^{-/-}$ hA3Tg mice ($n = 3$) and C57BL/6 mice ($n = 6$) were used to detect anti-GFP antibody response. Plasma of GFP reporter mice were used as comparison for natural anti-GFP Ab response: IL-4-GFP 4Get ($n = 8$)[72]; Mb1-GFP express GFP in B cells ($n = 6$)[73]; Nkx6.1 Venus Fusion transgenic mice express Venus, a yellow emission variant of GFP, in pancreatic β-cells (made by Ingo Burtscher, Munich) ($n = 9$), Foxo-1-GFP ($n = 8$)[74]; Blimp1-GFP express GFP in Plasma B cells ($n = 8$)[75]; CAT-GFP express GFP in Neurons, hair follicles and gut cells ($n = 7$). Serial dilutions of a pool of positive plasma were used as standard to normalize the assay. We used anti-mouse-IgG (H + L) HRP (Jackson Immuno Research) as detection antibody and o-Phenylenediamine dihydrochloride (Merck) as substrate. For total IgE measurements anti-IgE (clone R35-72, 10 µg/ml in 50 µl/well) was used to coat plates (Nunc) for detection goat anti-IgE (Southern Biotech, HRP labeled, 1:5000) was used. As standard mouse IgE (clone MEA-36, BioLegend) was used. The measurement was done with an Emax precision microplate reader (Molecular Devices).

For anti-DNA-IgG measurements (Fig. 6c) calf thymus DNA (Invitrogen, 10µg/ml in 50µl/well) was used to coat plates (Nunc). Half of the DNA had been incubated for 10 min at 100 °C, the other half was untreated. For detection goat anti-mouse IgG-POD (Thermo Fisher Scientific, 1:5000) was used. As standard serum from Plcg2$^{Ali5}$ mice (1:100) was used. The measurement was done with an Emax precision microplate reader (Molecular Devices).

For Western Blot detection of anti-(ERV)-GFP Ab response we used recombinant GFP (120 ng/ lane) (Abcam). Plasma from 2 to 4 months old EGT-315 B6 ($n = 11$); EGT-315 Tlr3$^{-/-}$Tlr7$^{-/-}$Tlr9$^{-/-}$ triple deficient mice ($n = 5$), EGT-315 Tlr7$^{-/-}$ mouse (genotype: Tlr3$^{+/-}$ Tlr7$^{-/-}$ Tlr9$^{+/-}$) ($n = 1$), EGT-315 Tlr7$^{+/-}$ mouse ($n = 1$), and C57BL/6 ($n = 5$) mice were used to detect anti-GFP antibody response. We applied anti-mouse-IgG (H + L) HRP (Jackson Immuno Research) as detection antibody, SuperSignal West Dura chemiluminescent substrate (Thermo Fisher Scientific) and a ChemiDoc XRS reader (BioRad) to detect the signals. For detection of Pcx by Western Blot we used 15 µg/ lane protein lysate of liver from C57BL/6 and EGT-315 B6 mice. Detection was done by anti-Pcx (1:1000; Thermo Fisher Scientific) and anti-goat IgG HRP (Thermo Fisher Scientific).

For ELISPOT assay, 100 ng GFP (Abcam) per well in 50 µl PBS was used to coat MAIPSWU10-plates (Merck-Millipore). They were incubated over night with 250.000 total spleen cells per well. FACS analysis of B cells was used to assure comparable percentage of B cells. After incubation, cells were removed, plates were washed, blocked and the anti-mouse-IgG (H + L) POX (Jackson Immuno Research) antibody and the ELISpot substrate AEC (Merck) was added. An Epson ELISPOT reader was used to quantify anti-GFP producing B cells.

## Cytometric bead array for mouse inflammatory cytokines

To determine serum inflammatory cytokine levels of C57BL/6, EGT-315 B6, Tlr3$^{-/-}$Tlr7$^{-/-}$Tlr9$^{-/-}$, EGT-315 Tlr3$^{-/-}$Tlr7$^{-/-}$Tlr9$^{-/-}$ mice we performed a BD Cytometric Bead Array (Mouse Inflammation and Th1-Th2 CBA, BD Biosciences, Germany). Serum samples were diluted according to the manufacturer's protocol. IL-12p70, TNF, IFN-γ, MCP-1, IL-10, IL-6, IL-5 and IL-4 APC-labeled beads were used together with PE-labeled detector reagent. The assay was measured at a FACSCalibur and analyzed via FlowJo10 software. Cytokine levels in serum correlate to the mean fluorescence intensity of each cytokine bead within the PE channel.

## PCR based screening of EGT-315 and identification of the ERV-GFP insertion

Genomic DNA was extracted from tail-biopsies and the primers 1384 5′-ACAACAATCTCACCTCTGACCA-3′ and 1385 5′-AAGTCGTGCTGCTTCA TGTG-3′ were used for identification of EGT-315 positive mice by generating a 549 bp product. For genotyping of hA3Tg mice the primers 5′-GGCACACAATGCCACACACTATGGCCTTCAGG-3′ and 5′-GTGCCCAGCATGTGTGCCATGGCTCAAGTTTG-3′ were used. Tlr7$^{-/-}$, Tlr3$^{-/-}$Tlr7$^{-/-}$Tlr9$^{-/-}$ mice were identified by PCR[11]. For identification of the ERV-GFP insertion which is based on the Mo-MuLV, we used a splinkerette based method[76]. We confirmed the insertion on Chr.19 by a PCR with Mo-MuLV specific primer and a primer from Chr.19 adjacent to the integration site 1694 5′-AACAGCTCCCACCTAGACAC-3′ and 1695 5′-GGAGACCCAGGGCTGTTAAT. Primers were designed using Primer3 software and PCR was done on a C1000 Cycler (Bio-Rad).

## Monoclonal antibody generation and in vitro inhibition assay

In order to test the function of antibodies against GFP from EGT-315 B6 mice we generated hybridomas by the technique of Köhler and Milstein[77]. Briefly, we used SP2/0 myeloma cells for the fusion and screened 200 proliferating clones. 12 ELISA-positive clones were further expanded and one clone No. 174 was purified and further characterized. The anti-GFP Ab was of the IgG2c isotype. Protein-G column-purified and biotinylated clone 174 anti-GFP antibody was tested by flow cytometry on Mo-MuLV pMOV-GFP infected NIH-3T3 cells using Streptavidin-APC (1:800, Thermo Fisher Scientific) for the staining. For the in vitro inhibition assay 5 supernatants from different anti-GFP hybridomas were used in 1:2 dilution, added to wild type WEHI-231 mouse B cell line co-cultured on a layer of NIH-3T3 cells expressing the pMOV-GFP Mo-MuLV. To test the inhibitory activity of sera from EGT-315 mice we also used this assay with serum diluted 1:14 in 215 µl volume on a monolayer of NIH-3T3 pMOV-GFP$^+$ cells and 100.000 WEHI-231 cells. After 18 h WEHI-231 were removed and counterstained with anti-CD19 PE (clone eBio1D3; Thermo Fisher Scientific) in order to eliminate NIH-3T3 CD19-negative cells from analysis for pMOV-GFP expression.

## hApobec3-induced mutations and Q-PCR quantification of ERV-GFP and B cell specific gene expression

To determine the efficacy of mouse and human Apobec3 deaminase activity ex vivo we isolated total RNA from spleen of EGT-315 C57BL/6 mice ($n = 4$), EGT-315 Tlr3$^{-/-}$Tlr7$^{-/-}$Tlr9$^{-/-}$ ($n = 5$) and EGT-315 Tlr3$^{-/-}$Tlr7$^{-/-}$Tlr9$^{-/-}$ hA3 ($n = 6$) (peqGOLD Total RNA Kit, VWR) and reverse transcribed the isolated RNA (RevertAid First Strand cDNA Synthesis Kit, Thermo Fisher). The GFP in the viral genome was then amplified by using GFP-specific primers 5′-ACAACAATCTCACCTCTGACCA-3′ and 5′-AAGTCGTGCTGCTTCATGTG-3′. The PCR products (549 nucleotides) were cloned into the pJET-2.1 vector (Thermo Fisher), and four to eight clones for each mouse were sequenced. Q-PCR was performed with the same cDNA used for sequencing using the primers 5′-CTA-CAATGAGCTGCGTGTGG-3′and 5′-CAAGCTCACACTTCATGATGG-3′ (actin, reference gene) and 5′-ACAACAATCTCACCTCTGACCA-3′ and 5′-AAGTCGTGCTGCTTCATGTG-3′ (ERV-GFP) to quantify ERV-GFP expression.

For B cell specific mRNA quantification splenic B cells were isolated by CD43 negative selection using MACS beads (Miltenyi). Total RNA was isolated using EXTRAzol (Blirt S.A.) and Chloroform ReagentPlus (SIGMA-ALDRICH). cDNA was synthesized with QuantiTect Reverse Transcription Kit (QIAGEN) using 1 µg of RNA template. For Q-PCR PowerUp SYBR Green Master Mix (Applied Biosystems), RNase-Free water (Invitrogen) and 10 pmol of forward and reverse primers were added to the cDNA template. Q-PCR-reaction was performed in Clear 8 tube strips (Bio-Rad) with optical flat 8-cap-strips (Bio-Rad) using the MiniOpticon RT PCR System (Bio-Rad). X-fold expression was calculated with the 2$^{-\Delta\Delta Ct}$-method. Primer: Actin - 974 b-

actin-FP-11    5′CTACAATGAGCTGCGTGTGG3′;    975    b-actin-RP-12-5′ CAAGCTCACACTTCATGATGG 3′; MuLV – 687 MLV1a- 5′ GGAGGGG-TACGTGGTTCTTT 3′; 688 MLV1b- 5′GCTGGACATCTTCCCAGTGT 3′; ERV-GFP - 1386-pMOV-GFP FW- 5′TATTCGGTTTACAGACGCCG3′; 1387 pMOV-GFP R- 5′CGTAGGTCAGGGTGGTCAC 3′; IFNg –1744 IFNg mouse qPCR-FW 5′CAGCAACAGCAAGGCGAAAAAGG3′; 1745 IFNg mouse qPCR-R 5′TTTCCGCTTCCTGAGGCTGGAT3′; STAT-1 – 1750 mStat1-1FP 5′GCCTCTCATTGTCACCGAAGAAC3′; 1751 mStat1-RP 5′TGGCTGAC GTTGGAGATCACCA3′; T-bet – 1760 mT-bet FP 5′CAACAACCCCT TTGCCAAAG3′; 1761 mT-bet RP 5′TCCCCCAAGCAGTTGACAGT3′; AID – 1754 mAID FP 5′TCTGCTACGTGGTGAAGAGGAG3′; 1755 mAID RP 5′ CCAGTCTGAGATGTAGCGTAGG3′.

## Reporter HEK-assays

Cells were seeded in 96-wells plates (HEK-Blue hTLR7 Cells (Invivo-Gen): 70,000 cells per well; 293-mTLR9-luc cells[70] 40,000 cells per well) and incubated for 24 h. Then they were stimulated in triplicates for 18 h with the supernatant of the stimulated B cells (C57BL/6 ($n = 3$)), Tlr3$^{-/-}$ Tlr7$^{-/-}$ Tlr9$^{-/-}$ ($n = 3$), EGT-315 B6 ($n = 1$) and EGT-315 Tlr3$^{-/-}$ Tlr7$^{-/-}$ Tlr9$^{-/-}$ ($n = 1$), with medium as a negative control and with R848 (1 µg/ml), and 1668 PTO-ODN (1µM, Phosphorothioate oligo-deoxynucleotide), respectively as a positive control. 20 µl supernatant of the stimulated HEK-Blue hTLR7 Cells were added to 180 µL QUANTI-Blue Solution (InvivoGen) per well. After 1 h incubation at 37 °C the optical density (OD) at 650 nm was measured using an Emax precision microplate reader (Molecular Devices). The HEK mTlr9 cells were lysed by adding 50 µl 1 x Reporter Lysis Buffer (Promega) and freezing (-80 °C) and thawing them (RT) four times. The firefly luciferase activity was measured using Berthold Detection Systems (Pforzheim, Germany).

## HEp-2 indirect immunofluorescence assay

The HEp-2 cells were brought into suspension using Trypsin and placed into chambers of multi-chamber slides (Corning). After incubating overnight, the cells were fixed with −20 °C cold acetone for 5 min and air-dried for 15 min. The cells were washed with PBS + 1% BSA, after which the mouse serum (1:50) was added and incubated for 1 h. Afterwards the serum was removed using PBS + 1% BSA and the cells were stained with Fc fragment specific anti-mouse IgG FITC (1:200, Thermo Fisher Scientific) and DAPI (1:10000, Thermo Fisher Scientific) for nuclear counterstaining. After 40 min of incubation the cells were washed with PBS + 1 % BSA and the chambers were removed from the slides, which were then mounted with Mowiol. Images were captured using a confocal laser scanning microscope (LEICA TCS SP5 II) and analyzed with Fiji software. An average of 19 cells per mouse were analyzed by marking the nucleus and measuring the mean fluorescence intensity of FITC and DAPI. The measurements were adjusted for background intensity, which was measured for each image separately in three different parts of the image.

## In vitro B cell proliferation/survival assays

Splenic B cells from C57BL/6, Tlr3$^{-/-}$Tlr7$^{-/-}$Tlr9$^{-/-}$, EGT-315 B6 and EGT-315 Tlr3$^{-/-}$Tlr7$^{-/-}$Tlr9$^{-/-}$ mice between 3 and 6 months of age were isolated to the purity of 93-98% by MACS beads (Miltenyi) against mouse CD43. CD43-negative B cells were seeded at a concentration of 10$^6$ cells/ml in 1 ml RPMI medium (10% FCS, P/S, β-ME). Cells were stimulated for 72 h with either LPS (10 µg/ml), anti-IgM (10 µg/ml, BioRad), R848 (1 µg/ml, InvivoGen), 1668 PTO-ODN (1 µM, Phosphorothioate oligo-deoxynucleotide, TIB MOLBIOL, Berlin), Pam3Cys (2 µg/ml, InvivoGen). In some experiments the Btk inhibitor Ibrutinib (10 µM, Selleckchem) was added. Propidium Iodide (2.5 µg/ml incubated for 30−60 s before measurement, BioLegend) was added before flow cytometry. In other experiments CellTrace Far Red was applied according to the manufacturer's instructions (Thermo Fischer). After stimulation for 72 h cells were analyzed with a FACSCalibur (BD).

## Histopathology

Kidneys were fixed in 4% PFA, routinely processed and embedded in paraffin. 4 μm sections were cut and dewaxed with xylene and rehydrated with a decreasing ethanol series. Hematoxylin and eosin staining (H&E) was performed with an autostainer (Microm, HMS 740, Dreieich, Germany) and periodic acid-Schiff staining (PAS) was carried out according to standard protocols.

## FISH: Cell culture, chromosome preparation and fluorescence in situ hybridization

Cells from each, a wild-type mouse C57BL/6 and mouse from strain EGT 315 B6 with insert were isolated; they were derived from skin of back, ear and tail as well as spleen. Those were taken into short term culture, and chromosomes were prepared following air drying method, according to standard protocols[78]. Chromosomes could be obtained for wild-type mouse C57BL/6 from spleen, and for strain EGT 315 B6 with insert from ear. The gene of interest was cloned in the pMOV-GFP plasmid; its DNA was isolated and labeled in SpectrumGreen by Nick translation. As a control a home-made whole chromosome painting (wcp) probe for mouse chromosome 19 labeled in Diethylaminocoumarine was used[79]. 5–10 metaphases were analyzed per sample.

## Whole-genome sequencing and data analysis

Genomic DNA from spleen of a single EGT-315 B6 was prepared and sent to Novogene UK. Whole genome sequencing was performed using a NovaSeq 6000 platform. The DNA was randomly sheared into short fragments. The obtained fragments were end repaired, A-tailed and further ligated with Illumina adapters. The fragments were size selected, PCR amplified, and purified. The library was checked with Qubit and real-time PCR for quantification and bioanalyzer for size distribution detection. The library was then sequenced. Reads were trimmed using Trim Galore (v0.6.4) and Cutadapt (v2.8)[80] in paired-end mode with default options. Illumina adapter sequences were removed. A quality Phred score cutoff of 20 was applied. Sequence pairs for which at least one read was shorter 20 bp were discarded. The resulting 350 mio reads (52 Bil bp) were mapped to *M. musculus* C57BL/6 J reference genome (NCBI RefSeq id GCF_000001635.27_GRCm39) and the pMOV-GFP plasmid sequence using segemehl (v0.3.4)[81] with split read mapping. The average sequencing depth was 31.07x with 96.64% of the genome covered at least 4x. Uniquely mapped split alignments predicted to be in trans with the pMOV-GFP plasmid were extracted (one part of the read was mapped to a chromosome, the other to the pMOV-GFP plasmid). These splits likely represent new pMOV-GFP insertions sites as they are not present in the genomic background of C57BL/6 J. 11 singular split reads that did not cover the 5' or 3' UTR of pMOV-GFP or could only be aligned to unplaced-scaffolds were discarded from the analysis. Partial integrations are either false-positives (e.g. due to high similarity of endogenous retrovirus sequences) or biological artefacts. A total of 20 splits with strong evidence were located in the library. Thereof, 19 in a region between positions 4,666,860 and 4,666,965 of chromosome 19 (Pcx gene) and 1 at position 116,835,161 of chromosome 13 (intergenic).

## Statistics and reproducibility

GraphPad PRISM 9 was used to analyze the results. They are presented as the mean of replicates with standard deviation (SD). No data were excluded from analysis. The samples for measurements were taken from individual mice. The samples for infected ES cells were tested in duplicates of the same sample (Fig. 1c). The number of biologically independent replicates is stated for each experiment. The unpaired student's *t*-test with Welch's correction was used for comparing two samples normally distributed. We used D'Agostino-Pearson normality test for evaluation of the distribution. For multiple samples normally distributed we applied two way ANOVA tests like Tukey's multiple comparisons test. For Q-PCR statistical anlysis we used unpaired Two-tailed student's *t*-test without Welch correction. Error bars, *P*-values are reported in figure legends.

## Reporting summary

Further information on research design is available in the Nature Portfolio Reporting Summary linked to this article.

## Data availability

The study generated two mouse lines EGT-315 and EZGT-332/3. In addition a number of monclonal antibodies against GFP have been generated. Mice and antibodies are available under standard MTA conditions of the Philipps Universität Marburg. Sequencing data of WGS are deposited at the NCBI Sequence Read Archive (SRA) under BioSample accession SAMN38798973. All other data are available in the article and its Supplementary files or from the corresponding author upon request. Any additional information required to reanalyze the data reported in this paper is available from the corresponding author. Source data are provided with this paper.

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

## Acknowledgements

We thank Axel Roers, Rayk Behrendt, Michael Bacher, Didier Trono, Hassan Jumaa and Lars Nitschke for their comments on the manuscript. We also thank Christian Möbs and Carolin Baum for help with the ELISpot assay, Thomas Wunderlich and Nadine Hövelmeyer for the kind and generous gift of ES cells. We are in debt for the generous provision of serum of GFP-reporter mice to Burkhard Schütz, David Vöhringer and Cornelia Symowski, Elias Hobeika, Andreas Diefenbach and Irene Mattiola, Wolfgang Schuh and Hans-Martin Jäck. Matthias Wabl and Hans-Martin Jäck supported us by providing the hA3 transgenic mouse line. Andreas Radbruch and Ute Hoffmann kindly provided us with T-bet deficient mice. We acknowledge Niklas Padutsch and Stefanie Kankel, Jena for excellent technical assistance with FISH. Thanks to Rolf Müller and Klaus Weber for the microinjection apparatus. Cornelia Exner was of existential help managing the animal experiment applications. Elisabeth Yu kindly did artistic rendering of the ERV-GFP virus. This work was supported by the DFG grant Yu 47/3-1 to P.Y. and grants BA 1618/7-1 (SPP1923 on Innate Sensing and Restriction of Retroviruses) and project A2 Transregio 237 (Nucleic Acid Immunity) to S.B. Open access funding provided by the Open Access Publishing Fund of Philipps-Universität Marburg.

## Author contributions

Conceptualization: P.Y.; methodology: E.R., T.A., S.D and S.P; formal analysis: E.R., T.A., M.Hu., A.L.K, M.L., F.B.L, A.Ka., T.L., A.R., P.D. and P.Y.; investigation: E.R., T.A., A.Ka., T.W., A.-C. K, M.Ho., K.K., A.L.K, A.Ku., C.K., T.L., F.B.L, V.L., A.R., S.G., P.D. and P.Y.; resources: M.B., I.B., S.K., T.W. and B.S.; writing-original draft: E.R., T.A. and P.Y.; writing-review & editing: M.S., S.B., T.A. and P.Y.; visualization: E.R., T.A., M.L., T.L., F.B.L, V.L., P.D. and P.Y.; supervision: M.B., M.Hu. and P.Y.; funding acquisition: S.B. and P.Y.

## Funding

## Competing interests

The authors declare no competing interests.
