## [Peer Review File · Nature Communications]

T-bet+ B cells are activated by and control endogenous retroviruses through TLR-dependent mechanismsREVIEWER COMMENTS

Reviewer #1 B cells (Remarks to the Author):

In this manuscript, Rauch et al proposed that ERV, which is incorporated into the genome can be continuously activated, can be eliminated by ERV-specific Tbet-positive B cells induced by TLR signals. By using a novel mouse line, EGT-315, the authors found that Tbet-positive B cells can be detected in ERV reactivation. This finding is interesting to consider the mechanism of Tbet-positive B cell formation. The fact that immune complexes of viral antigens and antiviral antibodies are deposited in glomeruli in kidneys of EGT-315 mice is also suggestive of the homology between ERV-induced immune responses and autoimmune reactions. Although these observations are potentially interesting, they are rather limited and descriptive.

1. They showed that EGT-315/TLR3/7/9 KO mice decrease anti-ERV-GFP antibody response, but the mechanism and biological significance are not obscure.
2. The authors emphasize that anti-ERV-GFP-specific B cells are important for immunosurveillance of ERVs, but there is little data to indicate that the germinal centre B cells and Tbet positive B cells induced in EGT-315 mice are anti-ERV specific B cells.
3. The authors used Tbet straight KO mice to demonstrate the importance of Tbet positive B cells in the immune response to ERV reactivation, but this does not address the importance of Tbet positive "B" cells. The involvement of Th1 cells in the anti-ERV immune response is also supposed to be important. They should analyze B cell-specific conditional Tbet KO mice or perform adoptive transfer experiments of Tbet-positive B cells.

Reviewer #2 ERV (Remarks to the Author):

The authors provided a model with a single unique ERV in the murine gremlin and used ERV-GFP expression to evaluate the immune response. I found the given results are very promising but only show a little concern about the genetic background of murine ERVs.

1. The author claimed that the virus inserted as a single copy, however the potential re-activation and re-integration the MuLV should be ruled out, such that whole-genome sequencing evidence should be provided.

2. Could the insertion of the provirus influence the expression of the neighboring gene, in this case the Pcx, that is irrelevant with the immune response?
3. The genetic background of other ERVs, similar to MuLV or close to and activated in murine cells, should be provided as immune responses sometimes could be cross-reactive.
4. The authors should also provide more convincing evidence that the ERV antigens have sole MuLV origin.

Reviewer #3 Tbet+ B cells (Remarks to the Author):

Rauch et al assess whether ERVs are recognized by the immune response as self or non-self antigen. They use a mouse model by which the GFP is encoded in the envelope of the ERVs, making the GFP a neoantigen (EGS-315 mice). They characterize these mice to show that the ERV-GFP+ cells are primarily found in immune related cells in the spleen and bone marrow. Moreover, they show that the incorporation of the GFP into the envelope specifically results in the production of anti-GFP IgG antibodies. The GFP used to drive this antibody response was only found on FDCs within the germinal centers of mice with intact TLR signaling. Furthermore, germinal center B cells in EGS-315 mice were not developed in the absence of TLR signaling. ERV signaling through TLRs lead to increased IFN γ expression which drove Tbet+ B cells that may produce the IgG2a/c anti-GFP. In the absence of TLR and anti-GFP antibodies the ERV response was not as well-controlled and found to be highly expressed in the intestine. The authors then show that the EGS-315 mice are capable of generating autoantibodies that lead to deposits in the kidney. Finally, the authors show that in the absence of TLR signaling in the EGS-315 mice there is an increase in non-specific IgE production. Overall, the authors find that impairment of TLR in B cell signaling results in the defective control of ERV with decreased IFN γ and IgG2a/c antibody responses.

The experiments are well thought out and address an important fundamental question of how the immune system recognizes ERVs. The data characterizing the ERV-GFP from their EGS-315 mice is strong. The work herein expands previous findings of the role of TLRs in recognition of ERVs. However, the lack of data identifying the Tbet+ B cell population, autoantibodies and IgE response lead to several overstated conclusions and require additional experiments or explanation to improve the novelty of their study.

Specific Comments:

The authors show Tbet expression in B cells via Tbx21 expression by qPCR and Tbet expression by FACS, but these data are not convincing that Tbet⁺ B cells are highly upregulated in EGS-315 mice. IFN- γ expression drives Tbet expression in both GC B cells and Age Associated B cells (ABCs), so examination of Tbet in B cells does not determine which population of B cells are increased. The authors show increase in GC B cells, so is the Tbet⁺ GC B cells or is there an increase in ABCs? Since these two B cells have distinct functional roles in the humoral response, identification of the Tbet expressing cells could be valuable to understand how the immune system controls ERVs.

The EGS-315 mice had elevated levels of anti-DNA autoantibodies, suggesting the response to ERVs are promoting this response. However, it was unclear to this reviewer how the EGS-315 had increased autoantibodies but reduced ANAs compared to B6 mice. Were there differences in anti-DNA titers or is this from a single dilution of sera? Furthermore, there was a clear presence of Ig deposits in the kidney, but whether this resulted in any type of kidney pathology such as proteinuria or cellular infiltrate was not assessed.

The rationale for the elevated IgE response in the absence of TLR signaling in the presence of ERVs identified in figure 7 is not clear. The idea that if IFN γ is not made then the response would switch to IL-4, but the authors do not show increased IL-4 or provide an explanation as to why IgE would be generated in the absence of GCs (as shown in Figure 4B).

Minor Comments:

Line 136, The author's state that Tlr7^{-/-} show about 62.5% ERV replication but the triple KO has 100%. Why do they think the role of TLR9 and 3 are in this process.

Fig 3b describes "ERV-GFP Medium" but there is no medium on the graph.

Fig 3D is not described in the text.

The markers used to identify the follicle in the top panel of Figure 4A are not described.

There is no quantification of the confocal data shown in Figure 4. Numbers of GCs per high power field containing GFP or some way to show that this is not a single GC that contains GFP in the EGT-315 mice.

The GC staining by FACS is not very convincing. Pre-gating on the IgDlo population may show a larger GC population in the EGT-315 mice. Furthermore, what is the IgDhi GL-7+ population circled in blue in Figure 4B, as GCs are IgDlo?

REVIEWER COMMENTS

Reviewer #1 B cells (Remarks to the Author):

In this manuscript, Rauch et al proposed that ERV, which is incorporated into the genome can be continuously activated, can be eliminated by ERV-specific T-bet-positive B cells induced by TLR signals. By using a novel mouse line, EGT-315, the authors found that T-bet-positive B cells can be detected in ERV reactivation. This finding is interesting to consider the mechanism of T-bet-positive B cell formation. The fact that immune complexes of viral antigens and antiviral antibodies are deposited in glomeruli in kidneys of EGT-315 mice is also suggestive of the homology between ERV-induced immune responses and autoimmune reactions. Although these observations are potentially interesting, they are rather limited and descriptive.

REPLY-1: With regard to the last sentence of reviewer #1 we would like to emphasize that we generated mouse strains that harbor traceable endogenous retroviruses. We further introduced three genetic deficiencies *in vivo* (human Apobec3G, nucleic-acid sensing TLRs, T-bet) to specifically analyze the expression and the immune response against ERVs. These loss-of-function models combined with ERV-GFP allowed us to study immune responses against ERVs in a mechanistic manner, and thus is not purely descriptive. Human endogenous retroviruses (HERVs) are implicated in a plethora of human diseases. However, research on (H)ERVs was limited by a lack of models to dissect ERV biology and the host response against ERVs. Our model addresses a mechanism how the self-encoded ERV is sensed and translated into a B cell response, namely T-bet⁺ ABC cells that are autoreactive by nature. This could extend our understanding of the recent finding that antibodies against ERVs promote lung cancer immunotherapy (Ng et al., 2023).

1. They showed that EGT-315/TLR3/7/9 KO mice decrease anti-ERV-GFP antibody response, but the mechanism and biological significance are not obscure.

REPLY-2: The central role of the ssRNA innate nucleic-acid sensor TLR7 in ERV immune control has been established (Jayewickreme et al., 2021; Young et al., 2012; Yu et al., 2012). The genome of endogenous retrovirus consists of ssRNA and therefore reactivation and expression does provide ligands for TLR7 mediated immune sensing. However, we have demonstrated that in single TLR7 knock-out mice on the C57BL/6 background genomic ERVs are not leading to 100% penetrance of ERV viremia. Only when TLR3 (Ligand: dsRNA) and TLR9 (Ligand: unmethylated CpG DNA) are also absent, as in the triple Tlr3^{-/-}Tlr7^{-/-}Tlr9^{-/-} mice a 100% penetrance of ERV viremia ensues. Even more importantly, only in the triple deficient mice an ERV mediated T cell leukemia develops. We have shown that TLR9 might be able to sense RNA-DNA hybrids generated during retrovirus replication (Obermann et al., 2019). The

mechanistic role of TLR3 in ERV control is indeed enigmatic to this date. This may imply an unexpected crosstalk or interdependency of all three NA-sensing TLRs in order to control ERVs which again highlights a mechanistic advance.

As to the biological significance we have identified that the absence of ABCs with increased T-bet expression might be the cause of lack of B cell response in (EGT-315-)Tlr3^{-/-}Tlr7^{-/-}Tlr9^{-/-} mice. An ever increasing set of data shows that these cells are important in chronic viral disease and in B cell autoimmunity (Mouat et al., 2022). Our model suggests that this comes together in the defense against ERVs explaining how it is even possible that a potential danger from within the genome can be targeted by the immune system without being impaired by self-tolerance mechanisms.

To further strengthen the biological significance of our model we have now analyzed the occurrence of spontaneous leukemia (T-ALL) induction. We previously showed that genome encoded ERVs are reactivated in Tlr3^{-/-}Tlr7^{-/-}Tlr9^{-/-} mice albeit with a late stage onset. The reason of the occurrence of T-ALL in these mice is based on activating oncogenes by insertional mutagenesis (Yu et al., 2012). In the submitted manuscript we could confirm this phenotype in EGT-315 Tlr3^{-/-}Tlr7^{-/-}Tlr9^{-/-} mice (Supplementary Fig. S1a, c). Here, we provide experimental evidence that T-bet expression regulation is essential for controlling T-ALL. At an age starting from 7 month more than 50% of EGT-315 T-bet^{-/-} succumb to leukemia, while only 38% of EGT-315 Tlr3^{-/-}Tlr7^{-/-}Tlr9^{-/-} and less than 10% of EGT-315 B6 are affected (REV-Figure 1a, 1b). Our additionally conducted experiments suggests that T-bet is indeed involved in the control of ERVs and their detrimental effects and can act as a tumor suppressor in the context of ERV-GFP-induced tumors.

We believe that the additional data provided here further strengthens our manuscript and advances mechanistic understanding regarding the connection of ERVs, T-bet and leukemia.

1a

1b

T-ALL tumor incidence

REV-Figure 1 | a, Examples of enlarged organs. b, EGT-315 Tbet^{-/-} mice succumb to T-ALL. Summary of mice analysed: EGT-315 B6 (3/34, 8.8%). EGT-315 Tlr3^{-/-}Tlr7^{-/-}Tlr9^{-/-} (5/13, 38%). EGT-315 Tbet^{-/-} (6/11, 55%). The mice examined were 7 month or older. Each dot represents a single mouse. For some of mice flow cytometry was done confirming aberrant T cell population (data not shown). *T-ALL cells from an EGT-315 Tlr3^{-/-}Tlr7^{-/-}Tlr9^{-/-} mouse show increased Tbet expression. **Two out of six lymphomas of EGT-315 Tbet^{-/-} mice were ERV-GFP positive.

2. The authors emphasize that anti-ERV-GFP-specific B cells are important for immunosurveillance of ERVs, but there is little data to indicate that the germinal center B cells and Tbet positive B cells induced in EGT-315 mice are anti-ERV specific B cells.

REPLY-3: We agree that for example a single cell analysis of anti-ERV-GFP specific B cells would be desirable to analyze their specific phenotype. However, isolation of these cells is technically difficult because they are rare and polyclonal (different anti-ERV-GFP BCRs). Their induction is dependent on spontaneous ERV-GFP activation as antigen source and therefore not synchronized compared to models that are characterized by exogenous antigen or virus injection. Although this is challenging we detected stages of antigen specific germinal center induction in spleen (revised Manuscript: Fig 4a and (newly added) 4b which corresponds to **REV-Figure 14**).

However, to provide an extensive analysis of the B cell compartment, we conducted additional experiments analyzing ERV-GFP specific B cells positive for Tbet. It is technically not possible to stain and sort for intracellular transcription factors and analyze viable cells. We tried to use surrogate surface markers for Tbet⁺ IgM memory B cells but in vitro experiments with isolated B cells from EGT-315 B6 mice using panning (immobilized GFP) and subsequent analysis were not successful (data not shown).

However, the data already contained in the manuscript and new experiments support our claims, namely

(A) the detection of the GFP signal from the deposition ERV-GFP virus in situ, in the light zone of the GC. We have added experimental data and quantitative analysis that show 7.5% of B cell follicles in younger and 2% in older EGT-315 B6 display ERV-GFP antigen deposition while none of 348 B cell follicles in spleen of EGT-315 Tlr3^{-/-}Tlr7^{-/-}Tlr9^{-/-} mice are positive (REV-Fig. 14).

and

(B), the increase of ERV-GFP expression and the reduction of anti-GFP antibodies in EGT-315 T-bet^{-/-} mice are strong indicators that the spontaneous and GFP Ag specific GC is a critical component of the ERV specific B cell response because it is absent in the TLR-deficient mice that do not produce Abs against ERVs. This adds another mechanistic insight since based on our data we can exclude an important role of extrafollicular responses and thus conclude that antibody-mediated ERV control is in fact GC-dependent. The significant suppression of anti-ERV-GFP antibodies in EGT-315 T-bet^{-/-} mice also strongly suggests that the important defect is in the B cell compartment. This notion is supported by the role of antibodies against human endogenous retrovirus in systemic lupus (Tokuyama et al., 2021).

3. The authors used T-bet straight KO mice to demonstrate the importance of T-bet positive B cells in the immune response to ERV reactivation, but this does not address the importance of T-bet positive "B" cells. The involvement of Th1 cells in the anti-ERV immune response is also supposed to be important. They should analyze B cell-specific conditional Tbet KOs or perform adoptive transfer experiments of Tbet-positive B cells.

REPLY-4: With the tools at hand we opted to perform adoptive transfer experiments. We tried to enrich T-bet⁺ B cells by negative selection of CD43⁻ CD23⁻ B cells from the spleen of EGT-315 B6 (donor) mice in a 2 step MACS isolation procedure. For a single transfer experiment a typical pool of 3 mice resulted in 150 Million splenocytes which allowed the isolation of an average of 50 million B cells. Second MACS negative selection by anti-CD23 reduced the population to between 850.000-1.3 Million CD23⁻ CD43⁻ B cells. The cells were then transferred by intravenous injection into EGT-315 T-bet^{-/-} recipients.

The rationale behind this strategy was:

a, EGT-315 B6 express anti-ERV-GFP specific T-bet⁺ memory B cells. b, a certain percentage might home to the spleen. c, encounter ERV-GFP antigen (which is present in the recipient) and d, then get activated expand or even lead to GC formation with ERV-GFP virus deposition and activation of specific antibody formation. This should lead to reduction of ERV-GFP expression. e, therefore, as readout we tested anti-GFP IgG and IgG2c titers, and in spleen cells ERV-GFP antigen expression and CD11c⁺T-bet⁺ B cells frequencies (REV-Fig. 2a, b). However, 4 weeks after transfer of T-bet⁺ enriched B cells we could not detect a significant increase in the parameters tested. Therefore, we think this approach is not conclusive probably due to the limited purity and insufficient numbers of T-bet⁺ B cells transferred (REV-Fig. 2c).

2a

T-bet MFI = expression von T bet in CD11c+ B cells:

Adop. Trans. of EGT-315 B6 B cells → EGT-315 T-bet^{-/-} mice

2b

Anti-GFP IgG in Adopt. Transf. mice

Anti-GFP IgG2c in Adopt. Transf. mice

2c

Exemplary flowcytometry to estimate the number of transferred ABCs

REV-Figure 2 | Adoptive transfer of enriched T-bet⁺ CD43⁻ CD23⁻ B cells from EGT-315 B6 to EGT-315 T-bet^{-/-} mice does not increase frequency of ABC B cells or titers of ERV-GFP specific IgG/IgG2c. a, Four weeks after adoptive transfer by i.v. injection of ca. 1 Million enriched B cells. Splenic ABC B cells were tested by flowcytometry. Dashed line indicates the background staining observed in T-bet-deficient mice. b, Four weeks after adoptive transfer serum concentrations of GFP specific IgG and IgG2c was tested and compared to the titers of untreated mice of different genotypes. c, Test of ABC enrichment method for adoptive transfer. Total spleen cells from three EGT-315 B6 mice were pooled and total spleen cells, total B cells (CD43⁻), the B cells that were positive for CD23 (CD23-bio and Streptavidin-MACS beads, Miltenyi) and bound to the MACS column and the unlabeled CD23⁻ B cells were stained with CD45RB-B220-FITC, CD11c-PE and intracellularly with T-bet-EF660 Abs. Right side, cell numbers of the populations. Dot plots gate for B220⁺CD11c⁺ ABCs. Histograms show T-bet⁺ cell populations percentages.

Indeed, the generation of a B cell specific T-bet^{-/-} mouse line with EGT-315 (e.g. CD19-Cre^{+/-}/Mb1-Cre^{+/-} T-bet^{-fl/fl} EGT-315^{+/-}) would provide desirable

information and would represent a more profound approach than the unsuccessful adoptive transfer experiments:

A, Intense inquiries suggest to me that at present no colleagues breed B cell specific T-bet^{fl/fl} mice. I asked John Wherry, Sebastien Storck, Gary Winslow, Rita Carsetti, Takahiro Adachi, Tomohiro Kurosaki and Steven Reiner (the creator of the T-bet^{fl/fl} mice) without success.

B, Therefore most recently, we have obtained Mb1-Cre mouse line (gifted by T. Winkler, Erlangen) and the mouse line: Tbx21flx (= T-bet^{fl/fl}) with IL17Acre x Rosa26YFP (gifted by C. Krebs, Hamburg Eppendorf). Mating 3 independent loci (Mb1-Cre^{+/-} T-bet^{fl/fl} EGT-315^{+/-}) and removing the unwanted genes IL17Acre x Rosa26YFP needs at least 3 generations of mice plus time for analysis of the phenotype. Also there is a regulatory affairs obstacle.

C, In Germany it is mandatory to apply for animal experiment permission, before mating or starting experiments with these mice because there is the chance that they will display a stressed/afflicted phenotype (German: belasteter Phänotyp) due to the potential T-ALL formation. This adds at least another 4-6 month to the timeline to obtain the permission from the responsible state authorities.

Therefore, we think that generating and analyzing the B cell specific T-bet deficient EGT-315 mice might need at least 1.4 years and may therefore be beyond the scope of this manuscript.

D, Nevertheless, we have listed 3 points (1.-3.) that support our notion that T-bet in B cells is very likely the crucial lymphocyte population in the control of ERV *in vivo*. Our own data and experiments from other labs make a key function of other cell types e.g. T cells in general or Th1-Tbet⁺ CD4-T cells unlikely (but not impossible).

1. Nucleic acid (NA) sensing TLR expression in T cells is in our opinion not functional (at least in murine models), while TLR7 and TLR9 function in B cells is well established and has been extensively studied.

In the past, we have also analyzed mice deficient for MyD88-, IRF-5 and Unc93b1 which all display ERV-viremia and lack of ERV specific B cells without a generalized defect in the T cell compartment (Yu et al., 2012) further supporting the notion of NA-TLRs not being relevant in T cells.

2. Data from the laboratory of G. Kassiotis (Crick Institute, London) demonstrated that B cell deficient mice but not T cell deficient mice have impaired ERV-control (Young et al., 2012). They showed that in the MHCII H2-A,E^{-/-}, Tcra^{-/-} and Tcrd^{-/-} mice (Young et al, Nature Fig. 1f) where no functional Th-1 T-bet⁺ T cells should be present no viremia has occurred.

3. In our submitted manuscript, we show that increased IFN- γ levels in serum are characteristic for the response in EGT-315 B6 mice but missing in EGT-315 Tlr3^{-/-}Tlr7^{-/-}Tlr9^{-/-} mice (Fig.5a). In order to test if an elevated Th1 cell response (which employs the T-bet transcription factor) is the source of the IFN- γ secretion we analyzed IFN- γ production *in vitro*. Here, we could not show that CD4⁺ and CD8⁺ T cells display statistical significant increase in IFN γ ⁺ T cell numbers (Suppl. Fig. S4b). This suggests that Th-1 T-bet⁺ T cells might not have an essential function in the control of ERV-GFP. Whether, NK cells or

other unknown cell types are dependent on T-bet and are the source of increased IFN- γ is of scientific interest but, in our opinion, beyond the scope of this manuscript.

Reviewer #2 ERV (Remarks to the Author):

The authors provided a model with a single unique ERV in the murine gremlin and used ERV-GFP expression to evaluate the immune response. I found the given results are very promising but only show a little concern about the genetic background of murine ERVs.

1. The author claimed that the virus inserted as a single copy, however the potential re-activation and re-integration the MuLV should be ruled out, such that whole-genome sequencing evidence should be provided.

Reply 5:

1. Our claim of a single copy integration of the ERV-GFP in the large intron of Pcx on Chromosom 19 is based on the iPCR technique we applied (Suppl. Fig. S2a and S2c). In particular the PCR in Fig. S2c demonstrates that the two primers

1694 EGT315-C19-FW	AACAGCTCCCACCTAGACAC
1695 EGT315-C19-RV	GGAGACCCAGGGCTGTTAAT

target the genomic sequence of Pcx (see *Mus musculus* strain C57BL/6J chromosome 19, GRCm39; NCBI Reference Sequence: NC_000085.7) at the integration site. After identification we confirmed the integration by PCR amplification from ERV-GFP and sequencing of the junction on the left and right side of the integration (data not shown).

2. We followed the suggestion of reviewer #2 and sent genomic DNA from spleen of a EGT-315 mouse for NGS Illumina sequencing (Novogene UK). The integrated ERV-GFP virus is based on Moloney-MuLV (pMOV-GFP (Sliva et al., 2004)) and therefore highly homologous to murine endogenous retrovirus sequences. Using a split read mapping approach, we were able to identify newly introduced MuLV sequences that were not already present in the C57BL/6 reference genome.

19 of 20 splits (new crossovers between 5' and 3' ends of MuLV and the target mouse genome locus) are located in a narrow region of chromosome 19. This confirms the original iPCR and FISH results suggesting a single germline integration of ERV-GFP. An additional ERV-GFP integration site identified at chromosome 13 is only supported by a single split. This might represent a somatic ERV-GFP insertion in spleen cells reflecting ongoing re-infection *in vivo*. (REV-Fig. 3a).

3. In addition Thomas Liehr at the University of Jena used the FISH technique to identify the chromosomal location of the integration(s) of the ERV-GFP. First he used the unique GFP-sequence of the ERV-GFP as a probe. This did not work due to the short length of the probe (716bp). The whole ERV-GFP plus cloning vector sequences (11kB) as a probe allowed enough signal strength to

identify that ERV-GFP has indeed integrated with a single integration site in chromosome 19 (REV-Fig. 3b).

a

b

REV-Figure 3 | a, Whole genome sequencing of EGT-315 revealed 20 splits, reads that contain either end of pMOV-GFP (Sliva et al., 2004) (= ERV-GFP) and the mouse genomic sequence adjacent to the insertion site. 19 overlapping splits (see Supplementary Table 3, manuscript) are located in a narrow region of 105 bps of chromosome 19 (upper part). This is the germline integration site. A single additional integration site was located in an intergenic region of chromosome 13 (lower part) and possibly represents a somatic insertion of the virus in a spleen cell(s). b, Upper panels, FISH analysis of chromosomes from a heterozygous EGT-315 B6 mouse hybridized with 11kB pMOV-GFP Plasmid DNA probe, containing the 8.3kB Mo-MuIV sequence and GFP (716bp) sequence. Upper right magnified positive chromosome 19 with ERV-GFP insertion located to subband 19B (insert green). Lower panels wild type control DNA from a C57BL/6 mouse.

4. In addition, the fact that in over 15 generations of mice we did not observe a change of inheritance frequencies according to Mendel's rules is an argument for a single autosomal integration of the ERV-GFP in EGT-315 mice. We found

that mating of heterozygous male EGT-315 B6 to C57BL/6 and heterozygous EGT-315 Tlr3^{-/-}Tlr7^{-/-}Tlr9^{-/-} to Tlr3^{-/-}Tlr7^{-/-}Tlr9^{-/-} resulted in the following genotypes of their offspring:

Inheritance of ERV-GFP

Total mice: 898 (100%)	EGT-315 B6 453 mice (50.4%)	wildtype 445 (49.6%)
Total mice: 576 (100%)	EGT-315 Tlr3 ^{-/-} Tlr7 ^{-/-} Tlr9 ^{-/-} 264 mice (45.8%)	Tlr3 ^{-/-} Tlr7 ^{-/-} Tlr9 ^{-/-} 312 mice (54.2%)

While it can't be excluded that very close to the integration on chromosome 19 a genetically linked copy of ERV-GFP might be present, it proves that no additional ERV-GFP integration occurred on a different chromosome. It also suggests that while ERV-GFP viremia present in EGT-315 Tlr3^{-/-}Tlr7^{-/-}Tlr9^{-/-} does lead to reintegration into somatic cells (causing T-ALL) it is not sufficient to penetrate the germline.

5. If the question 1. of reviewer #2 is related to re-activation and re-integration of the MuLV (or in our case ERV-GFP) in somatic cells the techniques of NGS or FISH are not applicable. From the experiments showing that in the serum of EGT-315 mice infectious ERV-GFP virus is present (Fig. 1f) it is clear that dividing cells, e.g. lymphocytes will be newly infected with re-integration of the virus. The identification of these random loci is very difficult. However, in 2012 we demonstrated that in T-ALL lines derived from leukemic Tlr3^{-/-}Tlr7^{-/-}Tlr9^{-/-} multiple new integrations can be identified, including in oncogenes (Yu et al., 2012). We assume that in EGT-315 Tlr3^{-/-}Tlr7^{-/-}Tlr9^{-/-} and in EGT-315 T-bet^{-/-} mice similar re-activation → lack of B cell mediated immunosurveillance → re-integration into the genome of somatic cells takes place. This would also explain the T-ALL formation in these two genetic backgrounds (REV-Figure 1a, 1b).

2. Could the insertion of the provirus influence the expression of the neighboring gene, in this case the Pcx, that is irrelevant with the immune response?

Reply 6 : This is an important aspect because Pcx has a role in anti-viral immune response by activating RIG-I viral dsRNA sensor (Cao et al., 2016). As noted we used EGT-315 heterozygous mice in all experiments. This would leave one normal allele of Pcx present in the mice used for experiments.

Nevertheless, we have added data to address the reviewer #2 point. Reverse transcribed quantitative PCR suggests that there is a decreased Pcx mRNA expression (REV-Fig. 4). However, this reduction is not statistically significant and does not result in significant reduced Pcx protein levels (REV-Fig. 4).

QPCR-Pcx

Western Blot Pcx

REV-Figure 4 | Left, RT-QPCR using liver mRNA from EGT-315 B6 and C57BL/6 mice. Right, western blot of protein lysate of liver with a anti-Pcx antibody and anti-actin antibody as control. We loaded 15 µg protein lysate on each lane. The signal strength of the Pcx POX enzyme was so strong that a visual signal could be observed directly on the membrane of the blot (Blot). We then stripped the buffer and reprobed the blot and measured a chemiluminescence signal at the expected size (upper part)

It is also interesting to note that Pcx expression in adult spleen is low, therefore we used liver protein extracts to study its expression (**REV-Fig. 5**).

REV-Figure 5 | NCBI Mouse encode transcriptome data for Pcx. Column 6 from the right depicts Pcx expression in in adult spleen tissue, column 12 expression in liver tissue.

Finally, since Pcx gene knockout mice are not viable and we can observe the existence of homozygous EGT-315 mice in our breeding in a normal Mendelian frequency we can exclude a complete suppression of Pcx expression due to the insertion of the EGT-315 allele in our mouse model.

3. The genetic background of other ERVs, similar to MuLV or close to and activated in murine cells, should be provided as immune responses sometimes could be cross-reactive.

REPLY-6: The exact genetic identity of the C57BL/6 derived ERV/MuLV is unknown. But we cloned and partially sequenced the ERV/MuLV reactivated in the TLR3^{-/-}TLR7^{-/-}TLR9^{-/-} mice on the C57BL/6 mouse genetic background. We and others concluded that the most likely candidate for the “resurrection” of ERV/MuLV in these mice is the EmV-2 locus (Jayewickreme et al., 2021; Yu et al., 2012).

However, in this manuscript we did generate the EGT-315 mouse line with the ERV-GFP fusion protein Env as a newly integrated germline endogenous retrovirus to avoid the question of where in the genome of the mouse the ERV/MuLV originated and what degrees of cross-reactive antibody we actually measure. To this end we analyzed in the majority of experiments the anti-GFP antibody response, because it is unique to the experimental ERV-GFP.

However, as we used the Molony-MuLV genome to generate the ERV-GFP it does have sequences that are closely related to the activated ERV/MuLVs of C57BL/6 mice. Therefore we can't exclude that the measurement of increase in ERV/MuLV expression we observed in EGT-315 mice might be due to cross reaction between the ERV(-GFP) part and ERV/MuLV. We have clarified this in the text of the manuscript (main text page 11: line 327 and 328).

4. The authors should also provide more convincing evidence that the ERV antigens have sole MuLV origin.

REPLY-7:

A, When we tested the specific immune response unique to the EGT-315 B6 line, we used antigen specific ELISA coated with GFP (commercially obtained). Neither wild type C57BL/6 mice nor various GFP-reporter mice harbor anti-GFP antibodies. In contrast, ERV-GFP carrying mice on the C57BL/6 background do express these antibodies. The genetic manipulation (Fig. 1a) and the iPCR results show that the GFP signal we observe in the EGT-315 and its antigenic property can only have their origin in the newly generated germline ERV-GFP.

B, When testing the ERV antigens of MuLV-type, which originate from the C57BL/6 murine genome. Example given, Fig. 6b. Here we used an antigen specific ELISA that uses whole MuLV virus isolated from a T-ALL line (Yu et al., 2012). We used an ultracentrifugation protocol to isolate the complete ERV/MuLV virus particle and tested the protein by western blot. We find the structural proteins of MuLV. In this publication we used electron microscopy, western blots, ELISA and flow cytometry (with an anti-MuLV-Env antibody (Evans et al., 1990), which all demonstrate that we have a *bona fide* MuLV. These preparations were used as ERV-antigens in the experiments described in our manuscript.

Reviewer #3 Tbet+ B cells (Remarks to the Author):

Rauch et al assess whether ERVs are recognized by the immune response as self or non-self antigen. They use a mouse model by which the GFP is

encoded in the envelope of the ERVs, making the GFP a neoantigen (EGS-315 mice). They characterize these mice to show that the ERV-GFP+ cells are primarily found in immune related cells in the spleen and bone marrow. Moreover, they show that the incorporation of the GFP into the envelope specifically results in the production of anti-GFP IgG antibodies. The GFP used to drive this antibody response was only found on FDCs within the germinal centers of mice with intact TLR signaling. Furthermore, germinal center B cells in EGS-315 mice were not developed in the absence of TLR signaling. ERV signaling through TLRs lead to increased IFN γ expression which drove Tbet+ B cells that may produce the IgG2a/c anti-GFP. In the absence of TLR and anti-GFP antibodies the ERV response was not as well-controlled and found to be highly expressed in the intestine. The authors then show that the EGS-315 mice are capable of generating autoantibodies that lead to deposits in the kidney. Finally, the authors show that in the absence of TLR signaling in the EGS-315 mice there is an increase in non-specific IgE production. Overall, the authors find that impairment of TLR in B cell signaling results in the defective control of ERV with decreased IFN γ and IgG2a/c antibody responses.

The experiments are well thought out and address an important fundamental question of how the immune system recognizes ERVs. The data characterizing the ERV-GFP from their EGS-315 mice is strong. The work herein expands previous findings of the role of TLRs in recognition of ERVs. However, the lack of data identifying the Tbet+ B cell population, autoantibodies and IgE response lead to several overstated conclusions and require additional experiments or explanation to improve the novelty of their study.

Specific Comments:

The authors show Tbet expression in B cells via Tbx21 expression by qPCR and Tbet expression by FACS, but these data are not convincing that Tbet+ B cells are highly upregulated in EGS-315 mice. IFN- γ expression drives Tbet expression in both GC B cells and Age Associated B cells (ABCs), so examination of Tbet in B cells does not determine which population of B cells are increased. The authors show increase in GC B cells, so is the Tbet the GC B cells or is there an increase in ABCs? Since these two B cells have distinct functional roles in the humoral response, identification of the Tbet expressing cells could be valuable to understand how the immune system controls ERVs.

Reply 8: To address this important point we adopted part of the in-depth analysis of flow cytometry panels recently established by the Shlomchik lab to characterize Age-associated B cells (ABCs) in the autoimmune MRL/lpr SLE model (Nickerson et al., 2023). The B cells that are increased in % T-bet+ cells (REV-Fig. 6) and MFI of T-bet (REV-Fig. 2a) were analyzed in the laboratory of Thomas Winkler, Erlangen and our laboratory in Marburg. Based on additional data generated, we conclude that splenic B cells expressing the markers CD19+, CD21/35-, CD11b+/-, CD23-, CD11c+ represent bona fide ABC cells. It is quite interesting that our model, which expresses ERV-GFP in the non-

autoimmune background C57BL/6, results in the very strong upregulation of T-bet⁺ ABC cells. The direct comparison to EGT-315 Tlr3^{-/-}Tlr7^{-/-}Tlr9^{-/-} mice and EGT-315 T-bet^{-/-} mice establishes that nucleic acid TLRs are indeed essential for the *in vivo* generation of these B cells. So, the mere insertion of a single copy of ERV-GFP into the genome of the non-autoimmune background C57BL/6 results in upregulation of ABCs.

REV-Figure 6 | ABCs in EGT-315 B6 mice. Gating strategy of CD19⁺ splenic B cells from C57BL/6 (n=1), EGT-315 B6 (n=3), EGT-315 TLR3^{-/-}TLR7^{-/-}TLR9^{-/-} (n=2) and EGT-315 T-bet^{-/-} mice (n=2). DN defines CD23⁻ CD21⁻ B cells. DP defines CD11c⁺CD11b⁺ B cells. Lower left, summary of mice tested for % T-bet positive DP cells.

With regard to GC B cells defined by the GL-7 expression we found that all DN B cells (CD21⁻/CD35⁻,CD23⁻) contain a distinct GC cell population whereas the ABC population is completely devoid of GL-7⁺ GC cells (**REV-Fig. 7**)

REV-Figure 7 | Overlay of histogram for GL7 of DN and T-bet⁺ CD11c⁺ ABC population of EGT-315 B6 mice (n=3). GL7 positive B cells are present in the DN B cell population but not in the ABC population in spleen of EGT-315 B6 mice. DN (mean= 9.48 SD= 0.627), T-bet⁺ CD11c⁺ ABC (mean= 1.14 SD= 0.752), ***P=0.0002. Unpaired t test with Welch's correction.

The EGS-315 mice had elevated levels of anti-DNA autoantibodies, suggesting the response to ERVs are promoting this response. However, it

was unclear to this reviewer how the EGS-315 had increased autoantibodies but reduced ANAs compared to B6 mice. Where there differences in anti-DNA titers or is this from a single dilution of sera? Furthermore, there was a clear presence of Ig deposits in the kidney, but whether this resulted in any type of kidney pathology such as proteinuria or cellular infiltrate was not assessed.

Reply 9: The reviewer pointed towards a discrepancy of increased anti-DNA autoantibodies measured by serum ELISA and reduced ANAs (in the Hep2 assay) in EGT-315 B6. It is important to note that DNA ELISAs measure antibodies specific to DNA, whereas HEp2 slides allow detection of autoantibodies binding to various structures within cells (ANA, cytoplasmic etc.). At this stage we can only speculate that in EGT-315 B6 the activated, replicating ERV-GFP virus induces DNA-specific B cells that preferentially recognize free double stranded or ssDNA. In the ELISA we use serial dilutions of individual sera and highly purified hering-sperm DNA as antigen. It is conceivable that in the HEp-2 assay (for this assay we did not titrate the sera) the target DNA is in a condensed chromosomal state in the nucleus of the cell and therefore potentially less accessible for the anti-DNA antibodies generated in the ERV-GFP expressing EGT-315 B6 mice. Further, DNA ELISAs display DNA in a highly accessible form simultaneously providing antigen complexes with an extremely high order of magnitude (due to being bound to the polystyrol surface).

In the HEp-2 assay we included sera from a severe autoimmune lupus model we studied in the past (Yu et al., 2005). We used this to bring the moderate autoimmune titers of EGT-315 B6 mice into the context of a lupus model that did suffer from glomerulonephritis caused by $Plcg2^{Al15}$.

Here, we add the analysis of the kidneys from EGT-315 B6. The mice do not progress to overt inflammation or chronic kidney pathology (**REV-Figure 8**).

REV-Figure 8 | Histopathological analysis of kidneys of different mouse strains revealed no definitive signs of glomerulonephritis. A, B: C57BL/6 (n=3); C, D: EGT-315 B6 (n=7); E, F: EGT-315 Tlr3^{-/-} Tlr7^{-/-} Tlr9^{-/-} (n=3); G, H: EGT-315 T-bet^{-/-} (n=2). Left panel (A, C, E, F): hematoxylin and eosin staining (H&E), right panel (B, D, F, H): periodic acid-Schiff. 200x magnification, scale bar: 100 μ m.

As mentioned in the submitted manuscript we think that this is due to a fundamental difference between an autoantigen and ERV-GFP virus antigens as an autoantigen. We think that ERV-GFP expression is suppressed by the ERV-GFP specific (auto-)antibodies. Therefore, an equilibrium of reactivation and suppression is established that does not necessarily cross a threshold to induce pathology but leads to constant life-long activation of autoreactive/ERV-GFP specific B cells. We assume that the autoantigenic-trigger in lupus models like Plcg2^{Ali5} or MRL/lpr is not eliminated and therefore makes these mice prone for immune complex mediated kidney pathology.

The rationale for the elevated IgE response in the absence of TLR signaling in the presence of ERVs identified in figure 7 is not clear. The idea that if IFN γ is not made then the response would switch to IL-4, but the authors do not show increased IL-4 or provide an explanation as to why IgE would be generated in the absence of GCs (as shown in Figure 4B). The rationale for the elevated IgE response in the absence of TLR signaling in the presence of ERVs identified in figure 7 is not clear. The idea that if IFN γ is not made then the response would switch to

IL-4, but the authors do not show increased IL-4 or provide an explanation as to why IgE would be generated in the absence of GCs (as shown in Figure 4B).

Reply 10: The cause of increased IgE response *in vivo* in ERV-GFP containing triple deficient EGT-315 is still not entirely clear:

1. Peng et al. showed that T-bet deficient B cells produced excess amounts of Th2-related isotypes IgG1 and IgE and they and others suggest that this did not simply result from an unopposed effect of IL-4 but further implicates T-bet in B cells in the direct regulation of IgE (and IgG1) (Peng et al., 2002).

Additionally generated data on the expression of T-bet in B cells demonstrates that the level of T-bet expression in EGT-315 Tlr3^{-/-}Tlr7^{-/-}Tlr9^{-/-} is nearly as low as in the EGT-315 T-bet^{-/-} mice (REV-Fig. 6, 2a). Therefore this nearly complete lack of T-bet in the EGT-315 Tlr3^{-/-}Tlr7^{-/-}Tlr9^{-/-} could explain increased ability to produce IgE *in vivo*.

2. As discussed in the submitted manuscript we could measure decreased IFN- γ levels in the IgE producing EGT-315 Tlr3^{-/-}Tlr7^{-/-}Tlr9 mice. In accordance to the hypothesis of Peng for T-bet^{-/-} a decrease of IFN- γ in our mice with unchanged but unopposed IL-4 levels could lead to increased switch to IgE.

Indeed, we add new data for measurement of IL-4 in serum and intracellularly in T cells. Both measurements are at the detection limits of the respective assays. For the serum cytokine array IL-4 expression is not enhanced in EGT-315 Tlr3^{-/-}Tlr7^{-/-}Tlr9^{-/-} mice. But surprisingly, is reduced compared to the EGT-315 B6 mice (REV-Fig. 9a). We hypothesize that the strong increase of IFN- γ in EGT-315 B6 mice (Fig.5b) might oppose the IL-4 effect *in vivo* in EGT-315 B6, but not in EGT-315 Tlr3^{-/-}Tlr7^{-/-}Tlr9 mice. Interestingly, the IL-4 levels of EGT-315 T-bet^{-/-} which lack Th-1 T cells are comparably low *in vivo* (REV-Fig. 9a). Measurement of the more stable Th-2 cytokine IL-5 in serum showed an slight although non-significant increase in EGT-315 Tlr3^{-/-}Tlr7^{-/-}Tlr9 mice (REV-Fig. 9b) as did the intracellular detection of IL-4 in *in vitro* stimulated T cells (REV-Fig. 10).

REV-Figure 9 | Serum cytokine bead array (BD) of serum probes of indicated mice. a, IL-4 b IL-5.

IL-4 *in vitro*

REV-Figure 10 | Isolated splenic CD4⁺ T cells were stimulated *in vitro* with PMA/Ionomycin. T cells were stained with intracellular flowcytometry for IL-4.

In summary the most likely explanation is that increased ERV-GFP virus expression in EGT-315 Tlr3^{-/-}Tlr7^{-/-}Tlr9^{-/-} mice is not sensed adequately and therefore reduction of IFN- γ results in extra germinal center activation of B cells through IL-4 present at comparable levels in the different mice. As stated in the submitted manuscript the increased IgE levels are not ERV-GFP specific. Therefore, we suggest that extrafollicular IgE induction or even a non-antigen specific interaction with basophils/mastcells (as IL-4 source) in bone marrow or other organs could lead to increased IgE even in the absence of GCs in EGT-315 Tlr3^{-/-}Tlr7^{-/-}Tlr9^{-/-} mice.

3. Finally, we tested the hypothesis discussed in the manuscript that direct infection or interaction of the ERV-GFP virus with B cells could trigger IgE class switch independent of CD40 signals or IL-4 (Morawetz et al., 1996; Yu et al., 1999). To achieve this we stimulated purified splenic B cells from C57BL/6 wild type and Tlr3^{-/-}Tlr7^{-/-}Tlr9^{-/-} mice for 4-5 days *in vitro* with purified ERV-GFP virus, Toll-like receptor ligands or as positive control in the presence of IL-4. In addition we used the 40LB system (Nojima et al., 2011) (for expansion and induction of GC-like B cells *in vitro*) with ERV-GFP present to test IgE induction. Neither ERV-GFP virus directly (**REV-Fig. 11**) nor 40LB cells expressing ERV-GFP were sufficient to induce IgE production in purified B cells from C57BL/6 or Tlr3^{-/-}Tlr7^{-/-}Tlr9^{-/-} deficient mice *in vitro* (**REV-Fig. 12**) Activation of IgE in EGT-315 Tlr3^{-/-}Tlr7^{-/-}Tlr9^{-/-} mice *in vivo* might not depend on a direct interaction with or infection of B cells by ERV-GFP.

REV-Figure 11 | Purified CD43⁺ splenic B cells from C57BL/6 (B6) and Tlr3^{-/-}Tlr7^{-/-}Tlr9^{-/-} mice were stimulated with purified ERV-GFP virus (50ng/ml) in DMEM medium in vitro for 4-5 days. As control B cells were stimulated with LPS (10μg/ml) with and w/o IL-4 (250U/ml). Only IL-4 and LPS induced IgE production which is absent in ERV-GFP stimulated cell supernatants (green boxes). Red dot depicts single EGT-315 B6 mouse tested.

REV-Figure 12 | B cells from C57BL/6 or Tlr3^{-/-}Tlr7^{-/-}Tlr9^{-/-} mice were co-cultured with 40LB wild type and 40LB-ERV-GFP (40LB expressing ERV-GFP), a B cells were harvested and stained with anti-CD45RB-B220 and gated for B cells. Green lines show individual B cells positive for ERV-GFP 72 h after start of co-culture. b Supernatant was taken after 4-5 days. In both genotypes presence of ERV-GFP⁺ 40LB cells (green boxes) did not induced IgE production. Addition of IL-4 induced IgE secretion in this system.

Minor Comments:

Line 136, The author's state that Tlr7^{-/-} show about 62.5% ERV

replication but the triple KO has 100%. Why do they think the role of TLR9 and 3 are in this process.

REPLY-11: ERVs as retroviruses in general induce synthesis of different nucleic-acids during replication. Their genome consist of ssRNA (two copies) and consistently Tlr7 does play a central role in ERV sensing (Yu et al., 2012) and control and retrovirus control respectively (Browne, 2011). However, Tlr7 single deficient mice do not display 100% penetrance of viremia and do not succumb to late onset T-ALL development (Yu et al., 2012).

At present redundancy in the TLR-system with regard to ERVs is not entirely understood, but we have shown that RNA-DNA hybrids which occur during retrovirus replication are an efficient ligand for Tlr9 (Obermann et al., 2019). Together with the reverse transcribed cDNA genome the ERV-GFP sensing of these ligands might induce activation of specific B cells and innate cells to mount an ERV suppressing B cell response. This could also explain the anti-DNA IgG measured (by ELISA) in EGT-315 B6 mice.

As to Tlr3 the data availability is even more scarce. Whether (murine) B cell express Tlr3 is not clear. Browne suggests that very little TLR3 is expressed in B cells (Browne, 2020) however in our hands murine B cells do not react to poly I:C (Yu, unpublished observation). Furthermore, the genome of ERV's is organized as ssRNA so only folding and complementary binding may generate regions of dsRNA. It is however not clear if this occurs and what contribution this postulated ligands do have in the ultimate control of ERVs *in vivo*.

Fig 3b describes "ERV-GFP Medium" but there is no medium on the graph.

REPLY-12: We apologize for this mistake. We have corrected the passage in the main text according to the Figure 3b legend (REV-Fig. 13).

227 Here, GFP was inserted into the proline-rich region of the Env gene generating a
228 fusion protein which is fluorescent while retaining infectivity of the virus (Fig. 1a, 1f). The
229 activation of the provirus DNA of EGT-315 results in deposition of the Env-GFP fusion
230 protein on the surface of cells (Fig.1a, 3d, S3b). After budding ERV-GFP virus particles
231 display Env-GFP on the the virus surface. This is probably the antigen that is
232 recognized by antigen specific B cells and leads to production of GFP specific IgG (Fig.
233 3a) while all other GFP-expressing reporter mouse lines were negative for anti-GFP
234 antibodies (Fig. 3a). Stratification of EGT-315 B6 mice into ERV-GFP high (>40%), low
235 (2-20%) and negative (0-0.5%) expressors shows inverse correlation of GFP expression
236 to the level of anti-GFP antibody titers (Fig. 3b). EGT-315 Tlr7^{-/-} mice and consistently
237 EGT-315 Tlr3^{-/-}Tlr7^{-/-}Tlr9^{-/-} mice expressed ERV-GFP but no anti-GFP antibodies (Fig.
238 3b-c, 3f).

REV-Figure 13 | Correction in the text according to the figure.

Fig 3D is not described in the text.

REPLY-13: On page 8 line 230 there is the reference to (Fig. 1a, 3d, S3b). The sentence starts at the end of line 228: The activation of the provirus DNA of EGT-315 results in deposition of the Env-GFP fusion protein on the surface of cells (Fig.1a, 3d, S3b).

The markers used to identify the follicle in the top panel of Figure 4A are not described. There is no quantification of the confocal data shown in Figure 4. Numbers of GCs per high power field containing GFP or some way to show that this is not a single GC that contains GFP in the EGT-315 mice.

REPLY-14: For the upper panels, ERV-GFP (green) and DAPI (blue) channels are shown. We have corrected this omission.

We completely agree with the point of reviewer #3. We now added data that includes quantification of GCs/ B cell follicles that contain ERV-GFP signals. We stained with PNA-Alexa 647 (GC), CD19-PE (B cells) and DAPI (Nuclei, in the exemplary photos we did not show this channel in order to make the specific signals more visible) and scanned the complete slides for ERV-GFP positive and negative B cell follicles in the highest magnification (63 objective). We counted in total 982 B cell follicles (combined of slides from a total of 31 individual mice) of the two different genotypes of interest (EGT-315 B6 and EGT-315 Tlr3^{-/-}Tlr7^{-/-}Tlr9^{-/-}) and ERV-GFP negative controls. We used confocal laser microscopy for the exemplary photos and also show a conventional immune fluorescence (IF) microscopy example of a spleen of an

EGT-315 B6 mouse. We summarized 3 independent experiments. Interestingly, we found in relatively young EGT-315 B6 mice (between 1- 2 month of age) a higher percentage of positive B cell follicles which suggests that at this stage the initial immune response against ERV-GFP starts (REV-Fig. 14). However, the presence of 2% of GFP⁺ B cell follicles in mice at 5 month of age shows that a constant re-emerging of ERV-GFP is sensed by the immune system.

REV-Figure 14 | ERV-GFP deposition in B cell follicles of EGT-315 B6 mice is absent in EGT-315 Tlr3^{-/-}Tlr7^{-/-}Tlr9^{-/-}. Left, quantification of ERV-GFP (green) vs ERV-GFP negative (black) B cell follicles of EGT-315 B6 young (positive 7.5%; 15+/186-; n=5; mean age = 1m19d), EGT-315 B6 old (2%; 6+/297-; n=12; mean age = 5m 24d), EGT-315 Tlr3^{-/-}Tlr7^{-/-}Tlr9^{-/-} (positive 0%; 0+/348-, n=10; mean age = 4m5d) and C57BL/6 and Tlr3^{-/-}Tlr7^{-/-}Tlr9^{-/-} (positive 0%; 0+/151-, n=4; mean age = 8m14d). Right, confocal microphotography representative of spleens stained with PNA-Alexa Fluor 647 (Peanut Agglutinin, red) and CD19-PE (blue). ERV-GFP is shown in green. Middle, microphotography of ERV-GFP deposition in standard immunofluorescence (IF) in the green channel.

The GC staining by FACS is not very convincing. Pre-gating on the IgDlo population may show a larger GC population in the EGT-315 mice. Furthermore, what is the IgDhi GL-7⁺ population circled in blue in Figure 4B, as GCs are IgDlo?

REPLY-15: As suggested by reviewer #3 we performed re-analysis of the existing data for IgD⁻ GL7⁺ B cells and could confirm that an increased number of GC B cells is present in EGT-315 B6 mice compared to the other genotypes (REV-Fig. 15).

REV-Figure 15 | IgD⁻GL7⁺ B cell frequency is increased in EGT-315 B6 mice.

Finally, the editor Timothy Powell and the reviewers requested us to further clarify how lack of nucleic acid (NA-) sensing TLR signaling contributes to the observed impairment of ERV control in our model.

We already discussed that two of the key signals for ABC Tbet⁺ B cell induction namely IFN- γ and Tlr7/Tlr9 (Cancro, 2020) are reduced or completely absent in NA- sensing TLR deficient mice respectively. In Fig.8 of the manuscript we showed data that IgM-BCR crosslinking (by anti-IgM) is not able to provide a signal to induce survival/proliferation in the absence of Tlr3, Tlr7 and Tlr9. In the supplementary Fig S6 we pinpointed this effect to Tlr7 rather than Tlr9 by stimulating B cells from the corresponding single Tlr-deficient mice. This points to a B cell intrinsic defect in Tlr3^{-/-}Tlr7^{-/-}Tlr9^{-/-} associated with BCR signaling irrespective of ERV-GFP expression. We also mentioned in the manuscript that the converse is true for B cells with a gain-of-function mutation in TLR7 in a situation of lupus (Brown et al., 2022).

Here we added an experiment we performed with the intention to analyze IgE production *in vitro* (REV-Figure 11). In the course of these experiments we stimulated purified B cells from C57BL/6 and Tlr3^{-/-}Tlr7^{-/-}Tlr9^{-/-} mice with LPS (Tlr4 ligand) and LPS+IL-4. After 72h we stained the B cells for CD23 expression (REV-Figure 16a, b). CD23 is expressed in mature follicular B cells and induced by IL-4 *in vitro*. A possible negative regulation of BCR signaling by CD23 has been suggested (Liu et al., 2016). The B cells from Tlr3^{-/-}Tlr7^{-/-}Tlr9^{-/-} mice have comparable CD23 levels w/o stimuli (red) but display dramatic upregulation of CD23 when only the Tlr4 ligand LPS (blue) is supplied. Also LPS+IL4 (green) lead to more CD23 induction when compared with B cells from C57BL/6 mice. Note that in C57BL/6 and IL4^{-/-}IL13^{-/-} mice about 50% of the B cells remain CD23^{low} while nearly all B cells of Tlr3^{-/-}Tlr7^{-/-}Tlr9^{-/-} mice become CD23^{high}. In contrast co-culture of B cells on 40LB with or without IL-4 or ERV-

GFP does not increase CD23 expression in $Tlr3^{-/-}Tlr7^{-/-}Tlr9^{-/-}$ B cells (REV-Figure 16c).

Red= w/o stimulus

Blue = LPS

Green= LPS+IL-4

b

c

REV-Figure 16 | LPS-mediated *in vitro* induction of CD23 is higher in Tlr3^{-/-}Tlr7^{-/-}Tlr9^{-/-} B cells compared to wild type B cells. a, Flowcytometry example of stimulated B cells. Stimulation of B cells from C57BL/6, Tlr3^{-/-}Tlr7^{-/-}Tlr9^{-/-} and IL4^{-/-} IL13^{-/-} mice for 72h with medium alone (red), LPS (10 µg/ml, blue) and LPS+IL-4 (IL-4 250U/ml, green). b, Summary of three independent experiments. Man Whitney U test pairwise comparison. ***P = 0.0007 and **P = 0.0031. B6 (n=10), Tlr3^{-/-}Tlr7^{-/-}Tlr9^{-/-} (n= 7). c, Coculture of B cells on 40LB with or without IL-4 or ERV-GFP (green cloumns) does not increase CD23 expression in Tlr3^{-/-}Tlr7^{-/-}Tlr9^{-/-} B cells compared to C57BL/6 B cells. C57BL/6 and Tlr3^{-/-}Tlr7^{-/-}Tlr9^{-/-} cells (black dots). EGT-315 B6 and EGT-315 Tlr3^{-/-}Tlr7^{-/-}Tlr9^{-/-} (red dots).

With this we add another piece of the puzzle as to why B cells of Tlr3^{-/-}Tlr7^{-/-}Tlr9^{-/-} mice are completely unable to produce anti-ERV antibodies. Tlr3^{-/-}Tlr7^{-/-}Tlr9^{-/-} B cells seem to have multiple defects: the genetic defect of nucleic-acid TLR-sensors leads to aberrant response to BCR and Tlr4 (LPS) activation. Whether increased CD23 induction *in vitro* reflects increased priming by IL-4 *in vivo* and leads to a postulated negative signal by CD23 (Liu et al., 2016) that impairs B cell function is unclear at this point.

References Revision:

- Brown, G.J., Canete, P.F., Wang, H., Medhavy, A., Bones, J., Roco, J.A., He, Y., Qin, Y., Cappello, J., Ellyard, J.I., et al. (2022). TLR7 gain-of-function genetic variation causes human lupus. *Nature*. 605(7909), 349-356. Published online 20220427 DOI: 10.1038/s41586-022-04642-z.
- Browne, E.P. (2011). Toll-like Receptor 7 Controls the Anti-Retroviral Germinal Center Response. *PLoS Pathog.* 7(10), e1002293. Published online 2011/10/15 DOI: 10.1371/journal.ppat.1002293
PPATHOGENS-D-11-01280 [pii].
- Browne, E.P. (2020). The Role of Toll-Like Receptors in Retroviral Infection. *Microorganisms*. 8(11). Published online 2020/11/19 DOI: 10.3390/microorganisms8111787.
- Cancro, M.P. (2020). Age-Associated B Cells. *Annu Rev Immunol.* 38, 315-340. Published online 20200127 DOI: 10.1146/annurev-immunol-092419-031130.
- Cao, Z., Zhou, Y., Zhu, S., Feng, J., Chen, X., Liu, S., Peng, N., Yang, X., Xu, G., and Zhu, Y. (2016). Pyruvate Carboxylase Activates the RIG-I-like Receptor-Mediated Antiviral Immune Response by Targeting the MAVS signalosome. *Sci Rep.* 6, 22002. Published online 20160224 DOI: 10.1038/srep22002.
- Evans, L.H., Morrison, R.P., Malik, F.G., Portis, J., and Britt, W.J. (1990). A neutralizable epitope common to the envelope glycoproteins of ecotropic, polytropic, xenotropic, and amphotropic murine leukemia viruses. *J Virol.* 64(12), 6176-6183. DOI: 10.1128/JVI.64.12.6176-6183.1990.
- Jayewickreme, R., Mao, T., Philbrick, W., Kong, Y., Treger, R.S., Lu, P., Rakib, T., Dong, H., Dang-Lawson, M., Guild, W.A., et al. (2021). Endogenous Retroviruses Provide Protection Against Vaginal HSV-2 Disease. *Front Immunol.* 12, 758721. Published online 20220104 DOI: 10.3389/fimmu.2021.758721.
- Liu, C., Richard, K., Wiggins, M., Zhu, X., Conrad, D.H., and Song, W. (2016). CD23 can negatively regulate B-cell receptor signaling. *Sci Rep.* 6, 25629. Published online 20160516 DOI: 10.1038/srep25629.
- Morawetz, R.A., Gabriele, L., Rizzo, L.V., Noben-Trauth, N., Kuhn, R., Rajewsky, K., Muller, W., Doherty, T.M., Finkelman, F., Coffman, R.L., et al. (1996). Interleukin (IL)-4-independent immunoglobulin class switch to immunoglobulin (Ig)E in the mouse. *J Exp Med.* 184(5), 1651-1661. Published online 1996/11/01 DOI: 10.1084/jem.184.5.1651.

Mouat, I.C., Goldberg, E., and Horwitz, M.S. (2022). Age-associated B cells in autoimmune diseases. *Cell Mol Life Sci.* 79(8), 402. Published online 20220707 DOI: 10.1007/s00018-022-04433-9.

Ng, K.W., Boumelha, J., Enfield, K.S.S., Almagro, J., Cha, H., Pich, O., Karasaki, T., Moore, D.A., Salgado, R., Sivakumar, M., et al. (2023). Antibodies against endogenous retroviruses promote lung cancer immunotherapy. *Nature.* 616(7957), 563-573. Published online 20230412 DOI: 10.1038/s41586-023-05771-9.

Nickerson, K.M., Smita, S., Hoehn, K.B., Marinov, A.D., Thomas, K.B., Kos, J.T., Yang, Y., Bastacky, S.I., Watson, C.T., Kleinstein, S.H., et al. (2023). Age-associated B cells are heterogeneous and dynamic drivers of autoimmunity in mice. *J Exp Med.* 220(5). Published online 20230224 DOI: 10.1084/jem.20221346.

Nojima, T., Haniuda, K., Moutai, T., Matsudaira, M., Mizokawa, S., Shiratori, I., Azuma, T., and Kitamura, D. (2011). In-vitro derived germinal centre B cells differentially generate memory B or plasma cells in vivo. *Nature communications.* 2, 465. DOI: 10.1038/ncomms1475.

Obermann, H.L., Eberhardt, I., Yu, P., Kaufmann, A., and Bauer, S. (2019). RNA-DNA hybrids and ssDNA differ in intracellular half-life and toll-like receptor 9 activation. *Immunobiology.* 224(6), 843-851. Published online 20190810 DOI: 10.1016/j.imbio.2019.08.001.

Peng, S.L., Szabo, S.J., and Glimcher, L.H. (2002). T-bet regulates IgG class switching and pathogenic autoantibody production. *Proc Natl Acad Sci U S A.* 99(8), 5545-5550. DOI: 10.1073/pnas.082114899.

Sliva, K., Erlwein, O., Bittner, A., and Schnierle, B.S. (2004). Murine leukemia virus (MLV) replication monitored with fluorescent proteins. *Virology.* 1, 14. DOI: 10.1186/1743-422X-1-14.

Tokuyama, M., Gunn, B.M., Venkataraman, A., Kong, Y., Kang, I., Rakib, T., Townsend, M.J., Costenbader, K.H., Alter, G., and Iwasaki, A. (2021). Antibodies against human endogenous retrovirus K102 envelope activate neutrophils in systemic lupus erythematosus. *J Exp Med.* 218(7). Published online 20210521 DOI: 10.1084/jem.20191766.

Young, G.R., Eksmond, U., Salcedo, R., Alexopoulou, L., Stoye, J.P., and Kassiotis, G. (2012). Resurrection of endogenous retroviruses in antibody-deficient mice. *Nature.* 491(7426), 774-778. Published online 2012/10/30 DOI: 10.1038/nature11599.

Yu, P., Constien, R., Dear, N., Katan, M., Hanke, P., Bunney, T.D., Kunder, S., Quintanilla-Martinez, L., Huffstadt, U., Schroder, A., et al. (2005). Autoimmunity and inflammation due to a gain-of-function mutation in phospholipase C gamma 2 that specifically increases external Ca²⁺ entry. *Immunity.* 22(4), 451-465. DOI: 10.1016/j.immuni.2005.01.018.

Yu, P., Lubben, W., Slomka, H., Gebler, J., Konert, M., Cai, C., Neubrandt, L., Prazeres da Costa, O., Paul, S., Dehnert, S., et al. (2012). Nucleic acid-sensing Toll-like receptors are essential for the control of endogenous retrovirus viremia and ERV-induced tumors. *Immunity.* 37(5), 867-879. Published online 2012/11/13 DOI: 10.1016/j.immuni.2012.07.018.

Yu, P., Morawetz, R.A., Chattopadhyay, S., Makino, M., Kishimoto, T., and Kikutani, H. (1999). CD40-deficient mice infected with the defective murine leukemia virus LP-BM5def do not develop murine AIDS but produce IgE and IgG1 in vivo. *Eur J Immunol.* 29(2), 615-625. DOI: 10.1002/(SICI)1521-4141(199902)29:02<615::AID-IMMU615>3.0.CO;2-I.

REVIEWERS' COMMENTS

Reviewer #1 (Remarks to the Author):

The authors have satisfactorily answered some of my queries. However, the ABC transfer part (REV-Figure 2) is very preliminary and the experimental design is not appropriate. The authors utilized CD43⁻ to isolate B cells, but in the published paper, ABCs are reported as CD43⁺. As a result, it seems that they are unable to enrich T-bet⁺ cells. Therefore, unless the authors can strengthen their findings, it would be appropriate to remove this part from the text. And, the title "T-bet⁺ B cell mediated immune surveillance" is not accurate and should be changed.

Minor

Fig5b: CD11c and CD11b FACS plots do not indicate double positive because they are not gated double positive.

Fig5d: IFN- γ should be "IFN-g"

Reviewer #2 (Remarks to the Author):

The authors have addressed my concerns.

Reviewer #3 (Remarks to the Author):

This is a revised manuscript from Rauch et al assessing whether TLR signaling by ERVs is impacting Tbet⁺ B cells (ABCs) to recognize these as a self or non-self antigen. The previous version of the manuscript had several issues leading to overstated and underdeveloped conclusions. The updated manuscript includes experiments that properly characterize the Tbet⁺ B cell populations, ABC and GC B cell in their EGS-315 mice. They had originally found anti-DNA antibodies in their model, they have further examined the pathogenic role of these autoantibodies in the EGS-315 mice by looking at kidney pathology. They have added several new figures examining how the cytokine production by T cells and the role of Tbet results in the elevated production of IgE in their mouse model. Furthermore, they have

added experiments that adequately address how the lack of nucleic acid sensing TLRs contributes to the impairment of ERV control in their model. I think the authors have satisfactorily addressed the reviewers concerns and I have no further comments.

Response to final revision NCOMMS-23-10084A

Reviewer #1 (Remarks to the Author):

The authors have satisfactorily answered some of my queries. However, the ABC transfer part (REV-Figure 2) is very preliminary and the experimental design is not appropriate. The authors utilized CD43⁻ to isolate B cells, but in the published paper, ABCs are reported as CD43⁺. As a result, it seems that they are unable to enrich Tbet⁺ cells. Therefore, unless the authors can strengthen their findings, it would be appropriate to remove this part from the text. And, the title "**Tbet⁺ B cell mediated immune surveillance**" is not accurate and should be changed.

REPLY-1: We concede to the criticism of reviewer #1 and remove REV-Figure 2 and the text relating to the adoptive transfer experiment.

Minor

Fig5b: CD11c and CD11b FACS plots do not indicate double positive because they are not gated double positive.

REPLY-2: We have changed the wording in the figure and thank the reviewer#1 for pointing out our mistake.

Fig5d: IFN- γ should be "IFN-g"

REPLY-3: We have changed the letter.

REPLY-4: We have changed the title to:

Tbet⁺ B cells are activated by and control endogenous retroviruses through TLR-dependent mechanisms

Reviewer #2 (Remarks to the Author):

The authors have addressed my concerns.

Reviewer #3 (Remarks to the Author):

This is a revised manuscript from Rauch et al assessing whether TLR signaling by ERVs is impacting Tbet⁺ B cells (ABCs) to recognize these as a self or non-self antigen. The previous version of the manuscript had several issues leading to overstated and underdeveloped conclusions. The updated manuscript includes experiments that properly characterize the Tbet⁺ B cell populations, ABC and GC B cell in their EGS-315 mice. They had originally found anti-DNA antibodies in their model, they have further examined the pathogenic role of these autoantibodies in the EGS-315 mice by looking at kidney pathology. They have added several new figures examining how the cytokine production by T cells and the role of Tbet results in the elevated production of IgE in their mouse model. Furthermore, they have added experiments that adequately address how the lack of nucleic acid sensing TLRs

contributes to the impairment of ERV control in their model. I think the authors have satisfactorily addressed the reviewers concerns and I have no further comments.